# Retroactive and graded prioritization of memory by reward

Erin Kendall Braun [1], G. Elliott Wimmer [2] & Daphna Shohamy[1,3]

Many decisions are based on an internal model of the world. Yet, how such a model is constructed from experience and represented in memory remains unknown. We test the hypothesis that reward shapes memory for sequences of events by retroactively prioritizing memory for objects as a function of their distance from reward. Human participants encountered neutral objects while exploring a series of mazes for reward. Across six data sets, we find that reward systematically modulates memory for neutral objects, retroactively prioritizing memory for objects closest to the reward. This effect of reward on memory emerges only after a 24-hour delay and is stronger for mazes followed by a longer rest interval, suggesting a role for post-reward replay and overnight consolidation, as predicted by neurobiological data in animals. These findings demonstrate that reward retroactively prioritizes memory along a sequential gradient, consistent with the role of memory in supporting adaptive decision-making.

[1] Department of Psychology, Columbia University, 406 Schermerhorn Hall, 1190 Amsterdam Ave MC 5501, New York, NY 10027, USA. [2] Max Planck University College London Centre for Computational Psychiatry and Ageing Research and Wellcome Centre for Human Neuroimaging, University College London, London WC1B 5EH, UK. [3] Zuckerman Mind Brain Behavior Institute and Kavli Institute for Brain Science, Columbia University, 3327 Broadway, New York, NY 10027, USA. Correspondence and requests for materials should be addressed to E.K.B. (email: eb2464@columbia.edu)

Decisions are shaped by memory for past experiences. There has been substantial progress in understanding the mechanisms by which the brain prioritizes memory for events that were themselves rewarding[1–4]. However, for memory to be useful in guiding decisions, it is essential not only to remember an event that was rewarding, but also—perhaps even more so—the sequence of events that led to it. Indeed, many experiences unfold over multiple steps before a reward is obtained —such as when a rodent explores a maze for reward or when a person explores a new city for a café. This poses an interesting problem, because in such situations, memory for the reward itself is not enough to know which steps are necessary to obtain the reward in the future. Computational models and physiological data offer possible solutions to this problem, but the implications for human memory have remained untested.

Computational models of goal-directed decision-making have proposed a family of theories, known as model-based reinforcement learning, which evaluates actions by integrating information from a learned model of the environment[5–8]. Memory is theorized to play a key role in such model-based behaviour because it offers a mechanism for encoding the associative structure of the environment, essentially offering an answer to the critical question of how the model in model-based decisions is constructed[9–13]. The hippocampus, known for its role in long-term episodic and spatial memory, is likely to contribute to the sort of structured associations that underlie the construction of a world model[1,9,13–17]. Moreover, theoretical work has suggested that the role of the hippocampus in building a world model may be specifically linked to its broader role in relational and temporal memories for individual events[18]. However, it remains unknown whether and how memory for sequences of events is shaped by reward in humans.

There is substantial evidence that reward prioritizes events for storage in long-term memory in both animals and humans. When the reward value of an upcoming event is known in advance, the beneficial effects of reward have been shown to be related to the anticipation of reward, before reward onset[2,3]. In many cases, however, the reward value is surprising and becomes known only afterwards. In such cases, effects of reward on memory must necessarily take place after encoding. Post-encoding and retroactive effects have been observed both in behaviour, with the demonstration that motivationally relevant outcomes can retroactively affect events that preceded them[19–21], as well as in the responses of hippocampal neurons following reward[22,23]. Following encoding, hippocampal neurons replay sequences of activity that reflect earlier encoding, a process which is thought to result in physiological strengthening of the memory trace[24–26]. Hippocampal replay often happens in reverse, beginning with the most recent events and playing back the trajectory from that point, rewinding the path that was taken. Crucially, rewards have been shown to increase the amount of reverse replay, suggesting a mechanism by which rewards could selectively strengthen memory for the preceding neutral events[22,23].

These data offer predictions about how reward could retroactively modulate sequences of events in humans and about which specific past events are likely to be prioritized in memory[27–30]. Hippocampal replay following maze navigation in rodents has been shown to occur concurrently with the firing of midbrain dopamine neurons, potentially allowing reward information to back-propagate to more distal spatial locations or decision points that preceded it[31]. This dopaminergic input may also selectively increase plasticity for the sequence of events that precede reward[27,32]. Together, these results suggest a mechanism by which reward could retroactively enhance memory for sequences of neutral events that preceded it and suggest that there should be a graded effect of reward on memory that would be strongest for those events closest in time and space to the reward itself.

The retroactive effects of reward on memory may require time for consolidation[19–21]. In humans, fMRI studies have shown that brain reactivation immediately following encoding predicts later long-term memory[33–35], and extensive research has demonstrated the importance of sleep for memory consolidation, specifically for motivationally relevant information[36–39]. Similarly, in animals, post-encoding replay has been shown in periods of quiescence immediately following exploration[22,23,40] and also during sleep[41,42], and experiments have shown that replay at both time points is necessary for learning[43–45]. Such findings suggest that the time window immediately after reward receipt (seconds) as well as the longer time window during sleep after learning (hours) may work jointly to prioritize sequences of events that lead to subsequent reward.

Collectively, these findings offer a unified framework that makes predictions about which particular events will be prioritized by reward and under which circumstances. Reward would be expected to prioritize memory for events in a sequence that are most proximal to the reward itself; these effects would be expected to depend on post-encoding processes and to emerge only after consolidation. Recent experiments in humans have demonstrated that motivationally relevant information modulates memory[2,46,47] and that post-encoding processes may also support memory modulation[4,19,20,33–35,37,39,48–50]. Yet, the critical prediction linking motivationally relevant outcomes and post-encoding consolidation to memory—specifically that reward will retroactively enhance memory for items as a function of their distance from the reward—has not been tested.

Guided by this framework, we tested three hypotheses regarding the effect of reward on memory during sequential goal-directed exploration in humans. First, we hypothesized that reward will selectively enhance memory for sequences of events leading to reward, as a function of their proximity to the outcome. Second, we hypothesized that the enhancing effects of reward on memory will depend on consolidation and therefore will emerge only after a delay of 24 h. Third, we hypothesized that retroactive effects of reward on memory will depend on processes occurring immediately after encoding and therefore predicted that the amount of time for rest immediately following sequences of exploration will modulate the effect of reward on memory.

To test these predictions, we developed a task, shown in Fig. 1, in which participants explored a series of grid mazes, one square at a time, searching for a hidden reward (a gold coin worth $1). Critically, during each navigational step, participants encountered an incidental, trial-unique object picture. To test the effect of reward on memory, we manipulated the maze outcomes so that half of the mazes ended in reward and half ended without reward. Later, we administered a surprise recognition memory test for the objects to examine whether the maze outcome (reward or no reward) retroactively modulated memory for the preceding objects. Importantly, during maze exploration, which was when objects were encountered, participants did not know whether that maze would end in reward or not. Thus, any reward-based modulation of memory must occur retroactively and cannot be due to differences in navigation or attention. To test the effect of consolidation on reward-modulated memory, some participants completed the memory test 24 h after encoding (24-hour condition), while others were tested 15 min after encoding (15-minute condition). Additionally, to test the effect of post-encoding processes immediately following encoding, we manipulated the amount of rest time (15, 20 or 25 s) following each maze. Finally, we ran three additional control experiments, designed to test the reliability of the results and to rule out alternative interpretations. Together, this series of studies reveals that reward systematically

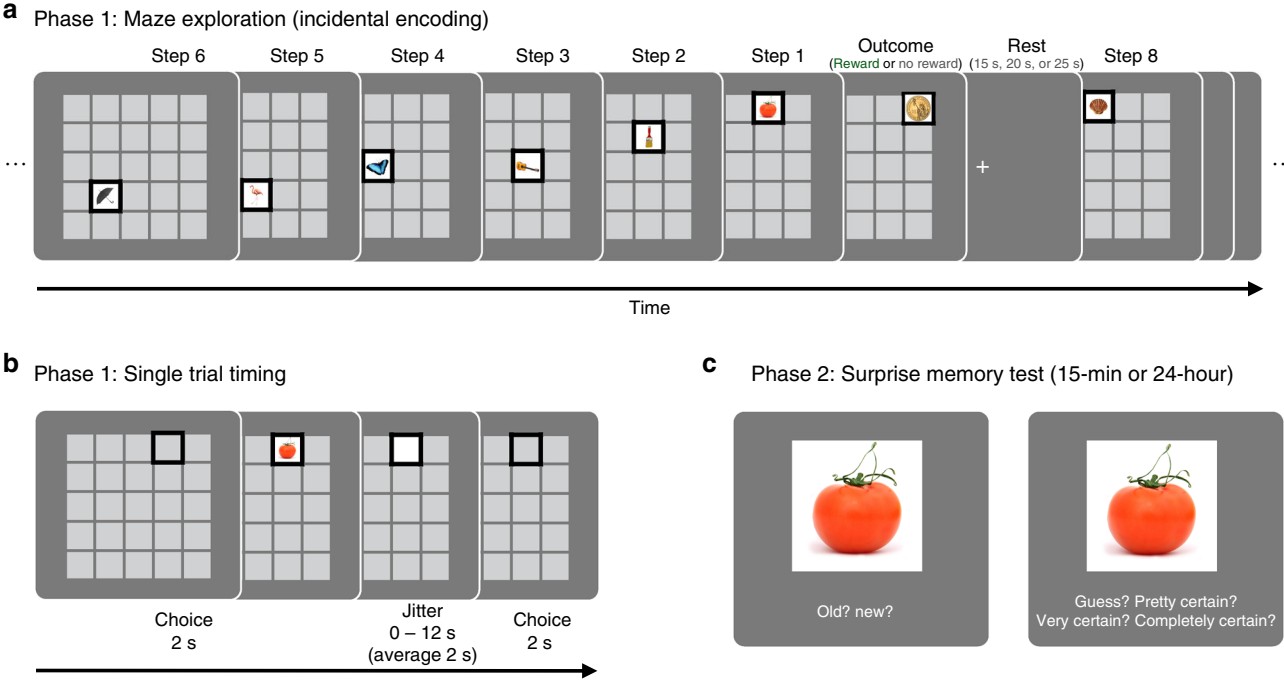

**Fig. 1** Experimental design to test the retroactive effect of reward on memory. The task consisted of maze exploration followed by a surprise memory test. **a** In the maze exploration phase, participants explored a series of mazes searching for a gold coin (each worth a bonus $1; 2007 Presidential $1 Coin image from the United States Mint). Mazes varied in length, ranging from 3 to 15 steps. Participants explored a total of 22 mazes. Half of the mazes ended in reward and half ended with no reward. During exploration, participants encountered trial-unique object pictures, appearing one at a time in the participants' current location. The objects were not related to the maze outcome and each object only appeared in one location within one maze during the entire task. The outcome of the maze was not known at the time the object was presented; therefore, any effect of reward on object memory must be due to retroactive modulation. A fixation interval of 15, 20 or 25 s followed each maze. During this time, participants were instructed to rest. **b** At each step, participants had 2 s to choose which adjacent square to navigate to next, and after taking that step, they saw a picture of an object appear in the chosen location for 2.5 s, after which the object was replaced with a white square for a brief interval. After that, the square turned grey, indicating that the participant should make his or her next navigational choice. **c** In the second phase, participants were given a surprise memory test after a delay of either 15 minutes or 24 hours. Participants were presented with a series of objects, one at a time, and were asked to indicate if the object was old (presented during the maze) or new (a lure) and the confidence of their response

prioritizes memory for neutral objects, retroactively enhancing memory for objects closest to the reward. This effect of reward on memory selectively emerges after a 24-hour delay and is stronger for mazes followed by a longer rest interval, suggesting a role for post-reward replay and overnight consolidation. These findings demonstrate that reward retroactively prioritizes memory along a sequential gradient, consistent with the role of memory in supporting adaptive decision-making.

## Results

**Retroactive prioritization of memory by reward after consolidation.** Our first hypothesis was that long-term memory would be prioritized for objects that were most proximal to the reward. Supporting this prediction, in Experiment 1, we found that when memory was tested after 24 hours of consolidation, reward had a retroactive and selective effect on memory as a function of proximity of the objects to reward (multi-level logistic regression; 24-hour condition ($n = 23$): reward × proximity: $\beta = -0.11$, SE $= 0.035$, CI$_{95} = [-0.18, -0.036]$, $p = 0.004$). Specifically, the closer an object was to the reward, the more likely it was to be remembered later, as shown in Fig. 2a (see also Supplementary Figure 1a, Supplementary Figure 2a,c, Supplementary Note 1). This interaction reflected a negative effect of proximity on memory in the rewarded mazes but not the non-rewarded mazes (multi-level logistic regression; proximity: reward mazes:

$\beta = -0.16$, SE $= 0.049$, CI$_{95} = [-0.25, -0.065]$, $p < 0.0004$; no-reward mazes: $\beta = -0.054$, SE $= 0.045$, CI$_{95} = [-0.32, 0.14]$, $p = 0.23$). Together, these findings indicate that the reward-driven reprioritization of memory varies as a function of the proximity of an object relative to the reward: the more proximal an object was to reward, the more likely it was to be remembered later.

In Experiment 2, we repeated the 24-hour condition of Experiment 1 in a new and separate sample of participants. The retroactive and graded effect of reward on memory was replicated (multi-level logistic regression; Experiment 2 ($n = 21$): reward × proximity: $\beta = -0.12$, SE $= 0.033$, CI$_{95} = [-0.18, -0.059]$, $p = 0.004$; proximity: reward mazes: $\beta = -0.15$, SE $= 0.048$, CI$_{95} = [-0.25, -0.050]$, $p = 0.00080$; no-reward mazes: $\beta = 0.098$, SE $= -0.052$, CI$_{95} = [-0.0053, 0.20]$, $p = 0.066$; see Supplementary Figure 1b and Supplementary Figure 3a, Supplementary Note 1, for post hoc tests see Supplementary Table 2). Experiment 3 sought to further replicate the reward proximity effect and to determine whether the result was affected by the location of the outcome. Specifically, in Experiments 1 and 2, on reward trials the gold coin was presented inside the maze, while on no-reward trials, the "maze over" outcome was presented outside of the maze. Consequently the reward proximity effect could simply have been caused by the outcome location and not reward per se. In Experiment 3, we repeated the 24-hour condition of Experiments 1 and 2, but controlled for the location

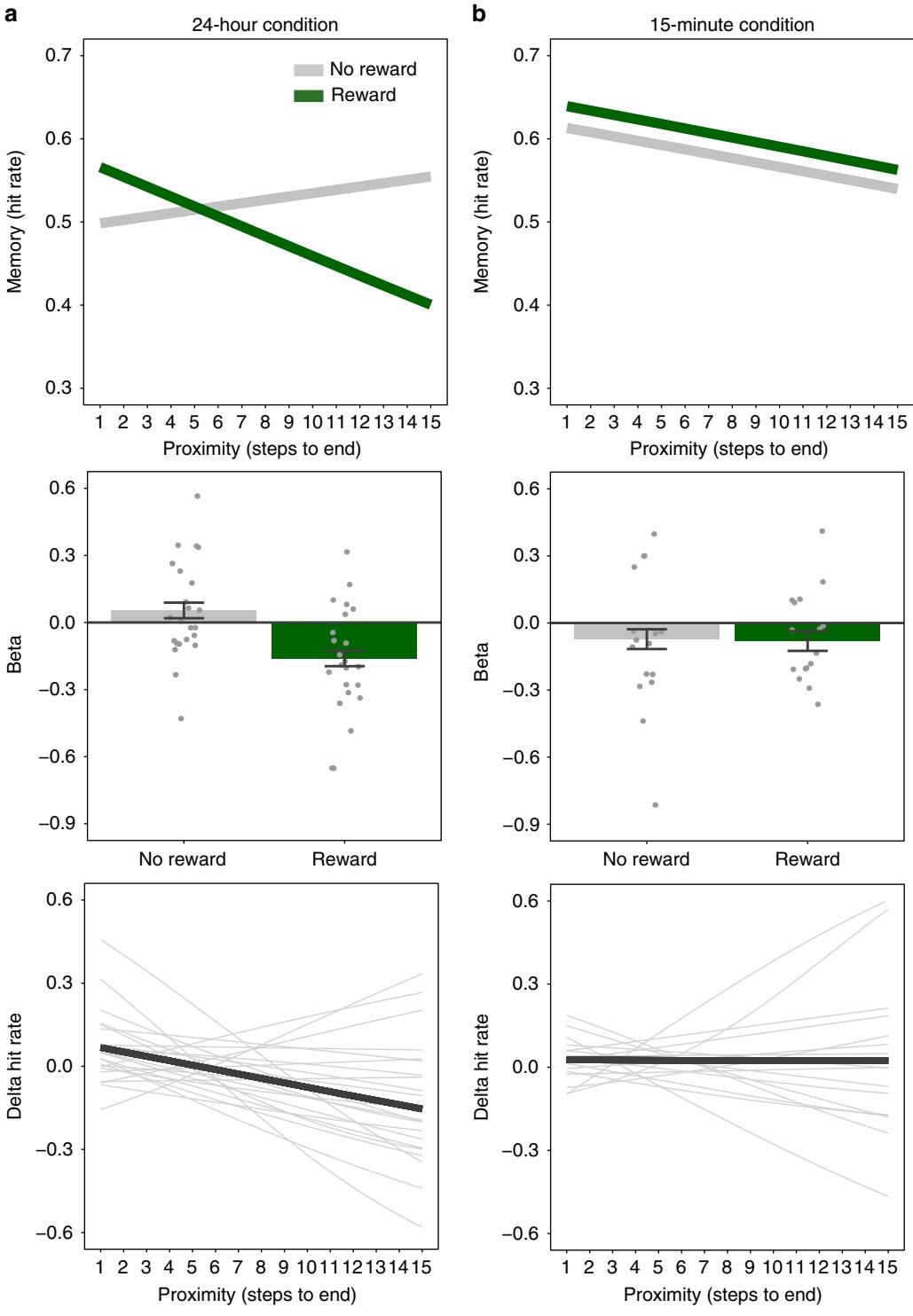

of the maze outcome: either a high reward (a gold coin worth $1) or low reward (a dime worth $0.10) was always presented in a maze square. When controlling for the location of the maze outcome, we again found the reward proximity effect (multi-level logistic regression; Experiment 3 ($n = 32$): reward × proximity: $\beta = -0.12$, SE = 0.027, CI$_{95}$ = [−0.17, −0.071], $p < 0.0004$; proximity: reward (gold coin) mazes: $\beta = -0.16$, SE = 0.038, z = −4.15, CI$_{95}$ = [−0.24, −0.085], $p < 0.0004$; low reward (dime) mazes: $\beta = 0.082$, SE = 0.038, z = 2.17, CI$_{95}$ = [0.0084, 0.16], $p = 0.030$; see Supplementary Figure 1c and Supplementary Figure 3b,

Supplementary Note 1, for post hoc tests see Supplementary Table 2). This indicates that the effect of reward on memory is not explained simply by differences in the maze outcome location but is due instead to retroactive reward modulation.

Importantly, this effect of reward and proximity on memory cannot be explained by simple primacy or recency effects (i.e. improved memory for the first or last items), as memory was modulated by the interaction of relative proximity and reward, rather than a main effect of either (see Supplementary Note 2). Additionally, the reward proximity effect is not confounded by

**Fig. 2** Selective retroactive modulation of memory by reward and proximity (Experiment 1). **a** 24-hour condition ($n = 23$). Rewards retroactively modulated memory, such that participants were more likely to remember objects that were more proximal to the reward. The top panel depicts the model predictions, showing how the proximity of the object was positively related to participants' memory for objects in rewarded vs. unrewarded mazes. The middle panel depicts the beta coefficients for the reward and no-reward conditions. The error bars represent the standard error of the reward × proximity interaction, and the dot plot overlay shows the reward and no-reward betas for each participant. The bottom panel depicts the interaction term representing the difference between the slopes in the reward and no-reward condition by proximity; the group level is shown in black, and individual participants are in light grey. **b** 15-minute condition ($n = 21$). For this condition there was no evidence for a reward by proximity interaction and a significant interaction with delay condition was observed, indicating a significantly greater reward proximity memory effect in the 24-hour condition. The top panel depicts the model predictions, showing the relationship between the proximity of the object and participants' memory for objects in rewarded vs. unrewarded mazes. The middle panel depicts the beta coefficients for the reward and no-reward conditions. The error bars represent the standard error of the reward × proximity interaction, and the dot plot overlay shows the reward and no-reward betas for each participants. The bottom panel depicts the interaction term representing the difference between the slopes in the reward and no-reward condition by proximity; the group level is shown with the black line and individual participants are in light grey

maze length, time elapsed between encoding and maze end, whether the reward stimulus was presented on the perimeter vs. inside of the maze, or the number of previous objects or rewards encoded in that maze square location (see Supplementary Note 2).

Our second hypothesis was that the effect of reward on memory would depend on consolidation. To test whether consolidation is necessary for the retroactive effect of reward on memory, we compared the data from participants who performed the memory test 24 hours after maze exploration with a separate group that performed the memory test 15 minutes after maze exploration. As shown in Fig. 2, the reward × proximity interaction in the 24-hour group was significantly greater than in the 15-minute group (multi-level logistic regression; delay condition × reward × proximity: $\beta = -0.061$, SE $= 0.024$, CI$_{95}$ $= [-0.11, -0.013]$, $p = 0.012$). Indeed, among those tested after 15 minutes, we found no interaction between reward and sequential proximity (multi-level logistic regression; 15-minute condition: reward × proximity: $\beta = -0.014$, SE $= 0.034$, CI$_{95} = [-0.051, 0.079]$, $p = 0.68$; proximity: reward mazes: $\beta = -0.039$, SE $= 0.048$, CI$_{95} = [-0.13, 0.058]$, $p = 0.41$; no-reward maze: $\beta = -0.067$, SE $= 0.054$, CI$_{95} = [-0.17, 0.033]$, $p = 0.19$; Fig. 2b, see also Supplementary Figure 1, Supplementary Figure 2b,d and Supplementary Note 2). These results support the hypothesis that the retroactive effects of reward on proximal experiences emerge selectively after consolidation.

**Reward proximity effect increases with longer rest intervals**. Our third hypothesis was that the reward proximity effect would depend on processes that occur immediately after encoding, and therefore, we predicted that the amount of rest immediately following sequences of exploration would modulate the reward proximity effect on memory. To test this hypothesis, we examined whether the duration of rest following incidental encoding in each maze modulated the effect of reward and proximity on memory. We manipulated the length of the post-encoding rest breaks to test if longer rest durations were associated with stronger retroactive reward modulation of memory for proximal objects. In the 24-hour condition, we found a significant interaction of proximity by reward by rest duration: for mazes followed by a longer break, the reward proximity effect was stronger (multi-level logistic regression; 24-hour condition: reward × proximity × rest duration: $\beta = -0.12$, SE $= 0.042$, CI$_{95} = [-0.20, -0.036]$, $p = 0.0048$; Fig. 3a). To further explore the effect of the rest duration interval in the delay condition, we compared model fits to the data. In a formal model comparison, we found that the model including rest duration explained the data significantly better than the simpler model omitting rest duration (chi-square test; $\chi^2(8) = 30.89$, $p = 0.014$). This effect was selective to the 24-hour condition (multi-level logistic regression; delay condition ×

reward × proximity × rest duration: $\beta = -0.067$, SE $= 0.031$, CI$_{95}$ $= [-0.13, -0.0054]$, $p = 0.033$); we found no effect of rest duration on the reward proximity effect among the participants tested after 15 minute (multi-level logistic regression; 15-minute condition ($n = 21$): reward × proximity × rest duration: $\beta = 0.012$, SE $= 0.050$, CI$_{95} = [-0.081, 0.12]$, $p = 0.74$; Fig. 3b). These results suggest that the rest breaks immediately following encoding are critical for the retroactive effects of reward on memory that emerge after consolidation.

These post-maze interval effects replicated in Experiment 2 (multi-level logistic regression; reward × proximity × rest duration: $\beta = -0.14$, SE $= 0.044$, CI$_{95} = [-0.23, -0.062]$, $p = 0.0016$; chi-square test; $\chi^2(8) = 33.50$, $p = 0.000050$; see Supplementary Figure 3c) and Experiment 3 (multi-level logistic regression; reward × proximity × rest duration: $\beta = -0.018$, SE $= 0.035$, $z = -5.06$, CI$_{95} = [-0.25, -0.11]$, $p < 0.0004$; chi-square test; $\chi^2(8) = 38.90$, $p = 0.0000051$; see Supplementary Figure 3d).

**Reward proximity effect is not due to strategic rehearsal**. Participants were not told of the memory tests before the maze exploration task. Nonetheless, it is possible that the effect of reward on memory could be related to strategic rehearsal or other explicit post-reward processes during the rest breaks. That is, an alternative explanation for the reward proximity effect is that during the rest intervals participants rehearse objects more when they are recent and from rewarded mazes. To test this possibility, we ran a follow-up experiment that investigated whether the opportunity to strategically rehearse during the post-encoding rest breaks is related to the reward proximity memory effect. This experiment was identical to the 24-hour condition, except that participants performed a secondary distractor task during the post-encoding rest interval, designed to prohibit strategic rehearsal during this time.

Participants were assigned to one of three conditions that varied the level of cognitive load of the distractor task (from low to high); this allowed us to test the effect of different levels of cognitive load during the rest interval on later memory. As shown in Fig. 4, one group of participants performed a target detection task, a second group of participants performed a spatial navigation task, and a third group of participants performed a working memory task. In all three groups, we replicated the main findings of an interaction of proximity and reward: we found a graded and retroactive effect of reward on memory after consolidation (multi-level logistic regression; target detection condition ($n = 27$): reward × proximity: $\beta = -0.14$, SE $= 0.029$, CI$_{95} = [-0.19, -0.082]$, $p < 0.004$; navigation condition ($n = 27$): reward × proximity: $\beta = -0.15$, SE $= 0.029$, CI$_{95} = [-0.21, -0.089]$, $p < 0.004$; working memory condition ($n = 23$): reward × proximity: $\beta = -0.13$, SE $= 0.032$, CI$_{95} = [-0.19, -0.064]$, $p < 0.004$, for post hoc tests see Supplementary Table 3,

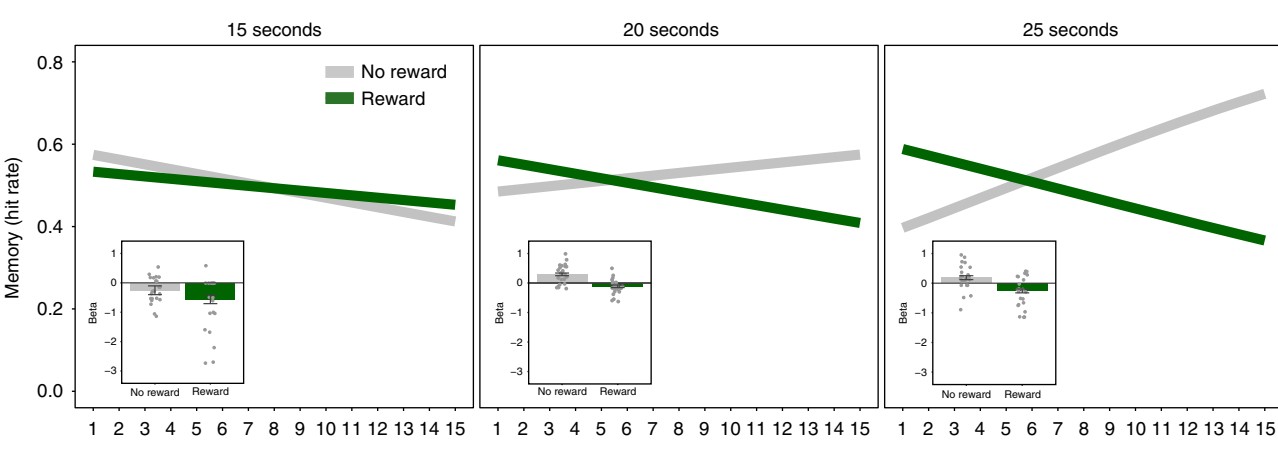

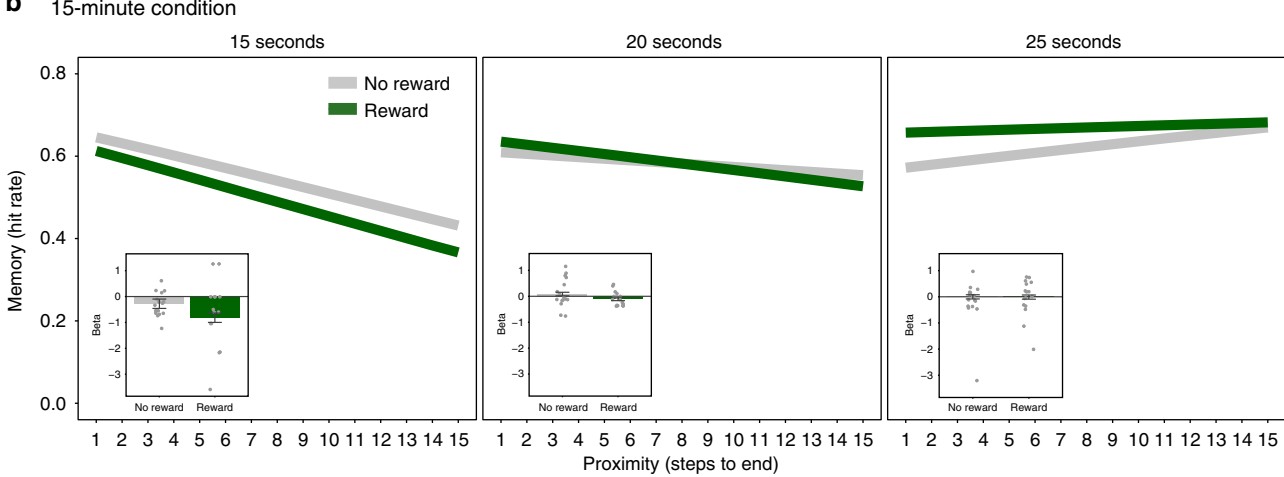

**Fig. 3** Reward proximity effect increases with longer post-encoding rest. (Experiment 1). **a** In the 24-hour condition ($n = 23$), we found that the duration of the rest break following each maze modulated the reward proximity effect, such that the interaction was stronger if the rest break following the maze was longer. **b** In the 15-minute condition ($n = 21$), we did not find an effect of the duration of the rest break modulating the reward by proximity interaction. The direct comparison of the 24-hour and 15-minute conditions showed a significant interaction (see Supplementary Table 1b for post hoc tests). The insets depict the beta coefficients for the reward and no-reward conditions. The error bars represent the standard error of the reward × proximity interaction, and the dot plot overlay shows the reward and no-reward betas across participants

see also Supplementary Note 1). The replication of this effect in all three conditions, even under challenging cognitive loads, suggests that the effect of reward on memory cannot be explained by rehearsal during these breaks.

Further, in all three conditions, we again found that the retroactive reward proximity effect was related to the duration of post-maze rest: specifically, the longer duration of the post-maze interval was related to a stronger reward proximity effect, irrespective of the distractor task condition (see Supplementary Figure 4). Additionally, again we found that the model including rest duration explained object memory performance better than the model without rest both in the target detection and working memory conditions (chi-square test; target detection: $\chi^2(8) = 39.86$, $p = 0.0000034$; working memory: $\chi^2(8) = 45.41$, $p = 0.00000031$) with a weaker effect in the same direction in the navigation condition (navigation: $\chi^2(8) = 13.57$, $p = 0.094$). These results replicate the previous finding that the duration of the post-maze rest period increases the reward by proximity interaction and demonstrate that the effect of rest duration on the reward proximity effect was not related to explicit rehearsal of objects

during rest, as the effect was robust to three different distractor tasks during the rest period.

**Retroactive prioritization of spatial memory by reward.** If rewards retroactively prioritize memory to facilitate future decisions, in addition to affecting memory for the object seen in each square, reward might also affect the location of each object, contributing to the formation of a cognitive map of the maze environment. To test the retroactive effect of reward on spatial memory, after participants completed the recognition memory test, we administered a surprise memory test for the objects' spatial location during memory encoding (Phase 3). In the spatial location memory test, an old object was randomly placed in the maze and the participant was instructed to move the object back to the square where the object was originally encoded (Fig. 5a). We then transformed the number of steps between the original (encoded) location and the remembered location into a continuous proportion correct measure (such that returning the object to the correct square was scored as a 1, and returning the

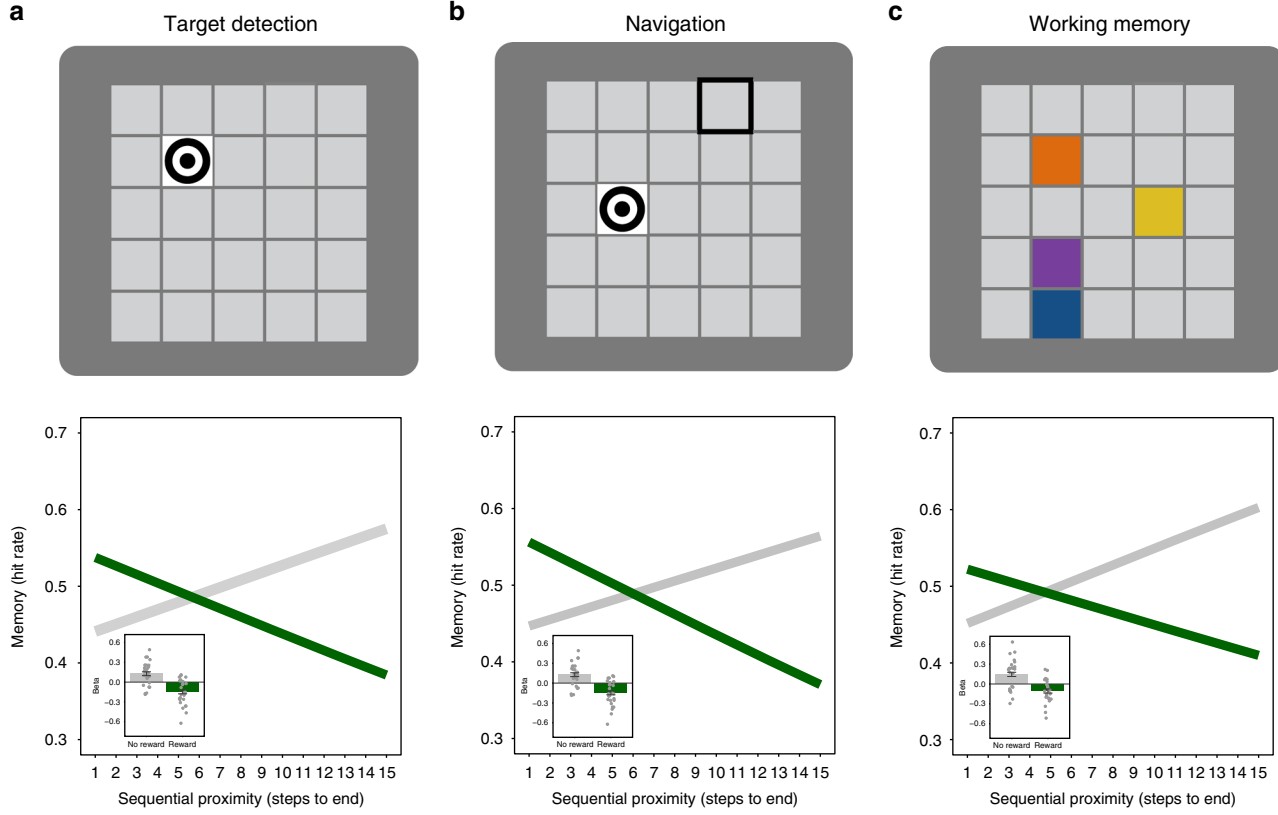

**Fig. 4** Reward proximity effect is not due to strategic rehearsal. (Experiment 3) To rule out the possibility that the reward proximity effect was due to strategic rehearsal during the rest intervals, we conducted a follow-up experiment in which participants completed one of three distractor tasks during the rest breaks. **a** In the target detection condition ($n = 27$), participants were instructed to make a key response every time a target image appeared in a maze square, but not when a lure image (a dark grey square) appeared (top). Despite the distractor task, we found that rewards retroactively modulated memory, such that participants were more likely to remember objects that were more proximal to the reward (bottom). **b** In the navigation condition ($n = 27$), participants used the arrow keys to navigate to a target (top). We again replicated the reward proximity effect. **c** In the working memory condition ($n = 23$), participants were presented with a target (a configuration of four randomly chosen colours in four randomly chosen squares) at the beginning of the rest interval and were instructed to make a key response to every presentation of this target configuration, but no responses to non-target configurations (other combinations of colours and squares) (top). In the working memory condition, we again replicated the reward proximity effect (bottom). We did not find a significant effect of distractor condition on the reward by proximity interaction (target detection condition vs. navigation condition × reward × proximity; target detection condition vs. working memory condition × reward × proximity (see Supplementary Table 3a for post hoc tests). The insets depict the beta coefficients for the reward and no-reward conditions. The error bars represent the standard error of the reward × proximity interaction and the dot plot overlay shows the reward and no-reward betas across participants

object to a square as far away as possible was scored as a 0, with intermediary values in between).

We found that participants' spatial memory performance was above chance (one-sample $t$ test; chance = 0.5; data from all Experiments combined: mean spatial memory (se) = 0.55 ± 0.00030, $t(145) = 13.35$, $p < 0.001$). Paralleling the results from the recognition memory test, we found that reward retroactively modulated spatial memory for sequentially proximal objects (multi-level logistic regression; data from Experiments 1, 2, 3 and 4, combined: reward × proximity: $\beta = -0.0062$, SE = 0.0022, $t = -2.83$, $CI_{95} = [-0.010, -0.0018]$, $p = 0.0032$; Fig. 5b), such that spatial location memory decreased as proximity to the end of the maze increased for the no (or low) reward mazes (multi-level logistic regression; Experiments 1–4 combined; 24-hour conditions only; $n = 146$: proximity—reward mazes: $\beta = -0.0047$, SE = 0.0032, $t = -1.48$; proximity—no (or low) reward mazes: $\beta = 0.0076$, SE = 0.0031; $t = 2.41$). Additionally, in this model, we detected a main effect of reward (multi-level logistic regression; data from all four experiments combined: $\beta = 0.0054$, SE = 0.0022, $t = 2.47$, $CI_{95} = [0.0011, 0.0095]$, $p = 0.0032$) such that spatial location memory for objects from rewarded mazes was

significantly better than in the non-rewarded mazes. We found qualitatively similar results when we repeated these analyses measuring the memory performance in steps instead of the proportion correct measure (see Supplement Note 3). While these effects are small, in combination with the parallel effects of reward on recognition memory, they suggest that spatial memory for the map of the environment is retroactively modulated by reward.

## Discussion

Together, these findings demonstrate that both recognition memory and spatial memory for neutral events encountered during goal-directed exploration are retroactively modulated by reward. The retroactive effect of reward on memory was graded, such that the objects closest to the reward were remembered best, and these effects only emerged after 24 hours. Moreover, this selective prioritization of memory by reward was positively modulated by the duration of brief rest periods immediately following exploration. These effects were replicated in six separate data sets in which we also demonstrated that the reward

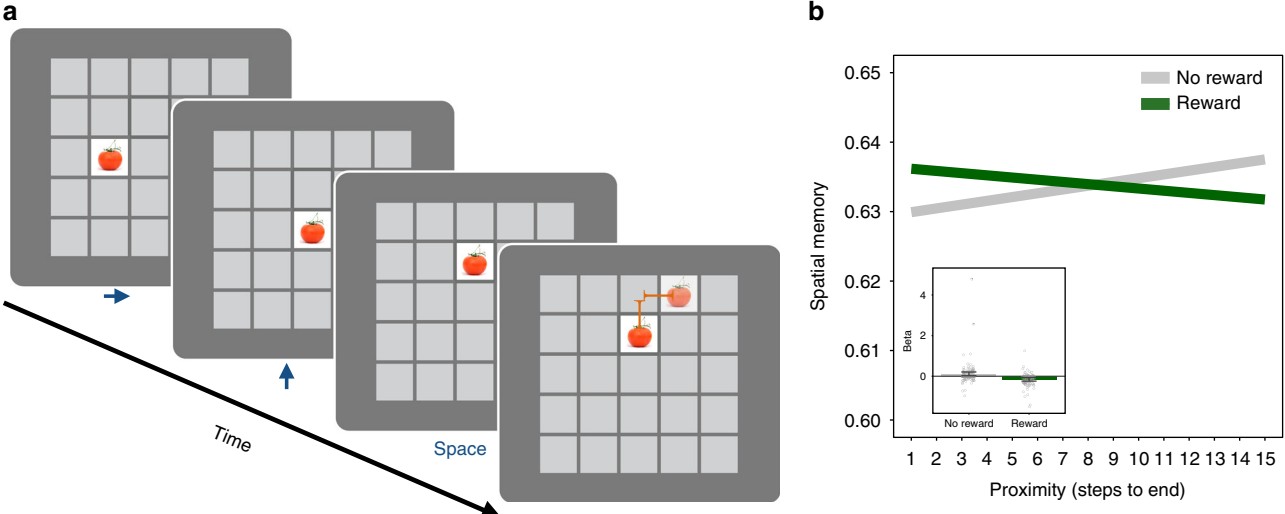

**Fig. 5** Rewards retroactively modulate spatial location memory. **a** Schematic of surprise spatial location memory test. An old object was randomly placed in a maze square, and participants were instructed to move the object back to the square where they originally saw the object by using arrow keys to move through the maze and then pressing the space bar to indicate the chosen location (self-paced). **b** Reward retroactively modulated spatial memory for sequentially proximal objects such that spatial location memory decreased as proximity to the end of the maze increased for the no (or low) reward mazes (Experiments 1–4 combined, 24-hour conditions only, $n = 146$). The inset depicts the beta coefficients for the reward and no-reward conditions. The error bars represent the standard error of the reward × proximity interaction and the dot plot overlay shows the reward and no-reward betas across participants

proximity effect on memory was not due the location of the maze outcome or to active rehearsal during the post-encoding periods.

Our finding that memory is biased by sequential distance from reward offers new insights into the mechanisms of both memory and decision-making. First, our findings offer an important link between goal-directed exploration, memory, and reward. Second, our findings are consistent with the idea that memories do not form a veridical representation of the world but instead are systematically modulated by motivationally significant events. In offering an explanation as to how memories are prioritized, our findings address fundamental questions in memory research about what we remember and why. Our results specifically demonstrate that objects that were closest to reward outcomes were retroactively prioritized in memory, rather than a general and immediate reprioritization of memory. This selective prioritization may help build world models that are well-suited to support adaptive behaviour by facilitating efficient, flexible, prospective choices in the future, such as finding a shortcut to get back to a rewarded location faster[28–30,51–57].

These results raise questions about the specific brain mechanisms that facilitate the retroactive reward proximity effect. It is well established that the neurotransmitter dopamine signals unexpected rewards[58] and other motivationally relevant events[59]. Dopamine is also known to facilitate synaptic long-term potentiation, possibly through midbrain-hippocampal circuits[1,60–63]. Our study further highlights the critical question of how unpredictable reward information could retroactively modulate memory traces for previously neutral information. In spatial navigation, hippocampal place cells replay the sequence of activity that unfolded during navigation during periods of rest that follow it[24,25,64,65], a process that is thought to facilitate memory consolidation[26]. Replay activity has been linked to reward in previous animal research in two ways: first, reward has been shown to increase reverse replay[23,40,66]. Second, replay immediately following receipt of a reward in a maze has been shown to occur concurrently with the firing of midbrain dopamine neurons, potentially providing a mechanism for reward information to retroactively affect traces of the preceding spatial locations or

decision points that preceded it[31]. Together, these findings suggest a physiological mechanism by which reward could retroactively enhance memory for preceding sequences of neutral events. Our findings provide behavioural evidence in humans that is consistent with this mechanism, demonstrating that reward has a graded effect on memory, enhancing memory for events closer in time and space to the reward itself[27].

Replay findings in rodents also suggest that retroactive effects of reward on memory may require time for consolidation before their effects manifest in behaviour. In rodents, replay has also been shown in periods of quiescence immediately following exploration[22,23,40] and also during sleep[67]. Studies suggest that replay at both time points is necessary for learning[43–45]. Therefore, the time window immediately after reward receipt (seconds) as well as the longer time window during sleep after learning (hours) may work jointly to prioritize sequences of events that led to subsequent reward. Our behavioural data are consistent with the hypothesis that replay that occurs immediately after encoding may tag events for later consolidation[61,63,68–70]. Although each replay event may occur very rapidly, questions remain regarding the necessary and sufficient duration of rest and sleep post encoding. Our data suggest that the duration of the time window post encoding has an effect on later memory, even on the order of several seconds (ranging from 15 to 25). Indeed, there is some evidence that both replay in rodents[23] and post-encoding reactivation in humans persist for prolonged durations following encoding[33–35].

Finally, in addition to finding that memory is prioritized for items close to the reward, we also found some evidence for deprioritized memory for objects that were closest to the no-reward (or low reward) outcomes—in Experiments 2, 3 and 4 and as a trend in Experiment 1. Although this finding was unexpected, it may be related to a recent report regarding the effects of reward on neuronal replay[23]. Ambrose et al. found that increasing or decreasing reward led to corresponding changes in the rate of reverse replay, while the rate of forward replay remained unchanged. To the extent that similar processes may be happening in our task, this may help explain why reward vs. no-

reward outcomes would predominately affect memory for events at the beginning of the maze, rather than only at maze end, leading to a within-maze primacy effect for the no-reward (or low reward) mazes. Further studies are needed to test this possibility directly.

In summary, our findings demonstrate that motivationally relevant events retroactively prioritize memories for preceding neutral experiences along a temporal-spatial gradient. As such, our findings provide important support for mechanisms of memory in service of decisions, and for how relational memory representations are modulated by reward to construct a world model that can support flexible, adaptive behaviour.

## Methods

**Experiment 1.** We designed Experiment 1 to answer three questions. (1) Do rewards exert a retroactive and graded effect on memory that varies as a function of the object's sequential proximity to reward? (2) Does the reward proximity effect depend on consolidation? and (3) Do longer post-encoding rest intervals result in a greater reward proximity effect? To address these questions, we designed a task in which participants explored a series of mazes, half of which ended in reward and half of which ended with no reward. Critically, at each step of the maze participants were incidentally presented with pictures of trial-unique objects; in this way, objects varied in their proximity to the end of the maze (either reward or no reward). Then, either 15 min or 24 h later, we gave a surprise recognition memory test for these objects. Since participants did not know the outcome of the maze at the time of incidental object encoding, any modulation of memory by reward is necessarily retroactive.

Participants: We recruited and tested participants from the Columbia University community in accordance with a protocol approved by Columbia University's Morningside Institutional Review Board. All participants provided informed consent. Participants were excluded if they reported any psychiatric diagnoses and/or use of psychoactive medications, had previously participated in a similar experiment, were non-compliant with the experimental protocol (e.g. did not show for the second session of the experiment), or if there were technical errors in data collection. Participants were compensated $12/h for their time and were paid a cash bonus based on the number of gold coins found (unbeknownst to the participants, this was always one gold coin during the practice and 11 gold coins during the task, for $12 total).

For Experiment 1, we recruited 51 participants. We excluded five participants due to technical errors, one participant for a self-reported psychiatric diagnosis and psychoactive medication, and one participant for previously participating in a similar task, leaving 44 participants (29 female, mean age (SD): 24.6 ± 6.3 years). Of these participants, 23 were assigned to the 24-hour condition, and 21 were assigned to the 15-minute condition.

Materials: We tested participants in individual testing rooms and presented the computerized tasks on a 21-inch iMac, using the Psychtoolbox (Brainard, D.H. (1997)) package in MATLAB. Participants indicated their responses on a standard keyboard. The experiment included a total of 288 unique object pictures: 192 objects displayed during the maze exploration task and an additional 96 objects presented as lures for the subsequent memory test. The gold coin stimulus was a picture of the Liberty side of a United States Zachary Taylor gold dollar (image from the United States Mint) presented on a white background.

Procedure: Participants completed the maze exploration incidental encoding task (Phase 1; Fig. 1a, b), a surprise recognition memory test (Phase 2; Fig. 1c), which began either 15 minutes or 24 hours after the maze exploration task and a surprise spatial location memory test (Phase 3, Fig. 5a).

Phase 1—Maze exploration (incidental encoding) task: The experimenter instructed participants that the goal of the task was to explore a series of mazes (5 × 5 grids) searching for a gold coin, each worth a $1 reward (see Supplementary Methods for instructions). Participants explored 22 mazes. Unbeknownst to them, we controlled the outcomes so that they found the gold coin in half of the mazes (11 reward mazes, 11 no-reward mazes).

Each maze began by displaying the 5 × 5 grid with all 25 locations filled with blank, grey-patterned squares. The participants' square location was indicated with a black frame. A square was randomly chosen for the participants' start location; no object was revealed in this start square. Then participants freely navigated to an adjacent square—up, down, left or right—by pressing the corresponding arrow key. This choice shifted the black location frame to the selected location and revealed an object in that square. Each object was presented for 2.5 s, after which we replaced the object with a white square for 0–12 s (jittered, mean 2 s). The next choice period was signalled by the square returning to the grey background colour, after which the participant had 2 s to choose where to move next. Participants were unaware that each maze had a predetermined length and outcome. After a participant explored the pre-allocated number of steps (i.e. squares explored or objects seen) for a given maze, he or she was presented with the maze outcome: for rewarded mazes, this was a gold coin; for non-rewarded mazes, "MAZE OVER" appeared at top of the screen. The outcome (the gold coin or "MAZE OVER") appeared for 2 s.

To create a subjective sense of exploration, we varied the number of objects presented within each maze, such that participants saw 3–15 unique objects per maze (mean: 8.7 objects), with lengths matched across the reward and no-reward conditions. In total, participants were presented with 192 objects during the maze exploration phase. We pseudo-randomized the order of objects, maze lengths and maze outcomes.

Each maze was followed by a 15, 20 or 25 s rest break, designated by a fixation cross. Participants were instructed to rest during this time. Following the rest break, the next maze began automatically.

If, within a particular maze, a participant re-explored a square, the same object was revealed so that the maze environment remained stable; however, we removed any object presented more than once from all analyses. If participants did not make a valid response within the allotted time (2 s), "Too Late" appeared at the top of the maze, and participants had to wait a full turn for their next move. Participants had a low rate of repeated objects (3.18 ± 3.29% of all trials on average) or "Too Late" missed trials (4.75 ± 5.04% of all trials on average). If participants tried to navigate outside of the maze, beyond the 5 × 5 grid, the response was not accepted.

At the beginning of the experiment, participants completed two practice mazes (one that ended with a gold coin and one that did not) with abstract shapes instead of objects. They were then paid a bonus $1 for the gold coin found during the practice, and the experimenter answered any questions.

After completing the maze exploration task, participants received an $11 bonus for the 11 gold coins found. Then, participants completed the tridimensional personality questionnaire (TPQ) so that the participants in the 15-minute condition would have a brief intermission between encoding and test. The results of the TPQ have not been analysed.

In the 15-minute condition, participants began the memory test after completing the TPQ (~15 min after the completion of encoding); in the 24-hour condition, participants began the memory test 24 h (± 2 h) from the beginning of the maze exploration (incidental encoding) task.

Phase 2—Surprise recognition memory test: In the second phase, we surprised participants with a recognition memory test for the object pictures incidentally presented during the maze exploration task. We presented the participant with 288 objects one at a time, consisting of all 192 old objects displayed during the maze exploration task and only 96 new lure objects to shorten duration of the memory test. On each trial, the object appeared for 1.5 s. Then, "Old" and "New" appeared on the bottom of the screen, under the picture, to prompt the participant to indicate his or her memory response, using the left and right arrow keys, respectively. After responding, we queried the participant's confidence, as is standard practice for subsequent memory tasks. The scale "guess (1) pretty certain (2) very certain (3) completely certain (4)" appeared at the bottom of the screen, and the participant indicated the confidence of his or her memory judgement using the 1 to 4 keys (see Supplementary Methods for instructions). Participant responses in the recognition memory test were self-paced.

Phase 3—Surprise spatial location memory test: In the third phase, we administered a surprise spatial location memory test to see if participants' memory for a given object's encoded location was retroactively modulated by reward. We presented an old object in an otherwise blank maze, and the participants' task was to move the object back to its encoded location. Participants were instructed that if they did not remember the exact square, they should move the object as close as possible to the encoded location to minimize the distance between the remembered location and actual encoded location (see Supplementary Methods for instructions).

On each trial, an old object appeared in a randomly chosen maze square; the encoded location was never chosen as the initial square. Then the participant used the arrow keys to move the object through the maze until the object was in the remembered maze square, which the participant indicated by pressing the space bar. After making a choice, the participant indicated his or her confidence on the scale "guess (1) pretty certain (2) very certain (3) completely certain (4)", which appeared at the bottom of the screen, using the 1 to 4 keys. The spatial location memory test was self-paced.

For Experiment 1, participants completed 60 spatial location memory trials. We randomly selected 60 objects that were correctly identified as old objects during the Phase 2: recognition memory test. If a participant did not have enough hit trials (in Experiment 1, this applied to two participants in the 24-hour condition), the balance of trials was filled with miss trials, and these trials were removed from subsequent analyses.

A subset of participants completed a reward memory test (results not reported). Then, each participant completed a written post-test questionnaire, which verified that the participant understood the task instructions and assessed their subjective experiences of the experiment, as well as a demographic form. Finally, participants were compensated for their time for all phases of the task.

**Experiment 2.** We conducted Experiment 2 to test if both the reward proximity effect and reward proximity modulation by rest duration results from the 24-hour condition of Experiment 1 replicated in a second sample. We repeated the exact same experiment in a second sample of participants. For Experiment 2, Phase 1 (maze exploration incidental encoding task) and Phase 2 (surprise recognition memory test) were identical to Experiment 1; Phase 3 (surprise spatial location memory test) was the same as Experiment 1, except that participants completed 90 spatial location memory trials, instead of 60.

In Experiment 2, participants had a low rate of repeated objects (2.42 ± 3.04% of all trials on average) or "Too Late" missed trials (4.61 ± 5.36% of all trials on average).

Participants: For Experiment 2, we recruited 25 participants. We excluded two participants due to self-reported psychiatric diagnoses and/or psychoactive medication, one participant for not showing up to the second session (i.e. the memory test), and one participant for refusing to turn off his mobile phone during the maze encoding task, resulting in 21 participants (12 female; mean age (SD): 25.7 ± 5.1 years). Due to technical errors, only 20 participants completed the Phase 3: surprise spatial location memory test.

**Experiment 3**. In Experiments 1 and 2, the reward maze outcome (i.e. gold coin) was always presented in a maze square and the no-reward outcome (i.e. maze over) was always presented above the maze. In these experiments we cannot disambiguate whether the reward proximity effect was caused by reward or by the location of the maze outcome, and consequently, we conducted Experiment 3 to test whether the reward proximity effect was due to reward modulation by comparing high vs. low rewards, always presented in the last square of the maze. In Experiment 3, participants had a low rate of repeated objects (2.34 ± 3.89% of all trials on average) or "Too Late" missed trials (6.13 ± 2.57% of all trials on average).

Participants: For Experiment 3, we recruited 42 participants. We excluded one participant due to technical error, six participants due to self-reported psychiatric diagnoses and/or psychoactive medication, one participant for not showing up to the second session (i.e. the memory tests), one participant for previously participating in a similar task, and one participant for misrepresenting his or her age (older than 35-years-old), resulting in 32 participants (20 female; mean age (SD): 24.0 ± 4.7 years). Due to technical errors, only 28 participants completed the Phase 3 surprise spatial location memory test. For Experiment 3, participants were compensated $12/h for their time and $13.20 bonus ($1.10 for the practice and $12.10 for the maze exploration encoding task).

Materials: For Experiment 3, the stimuli were identical to the stimuli used in Experiment 1, except that participants were additionally presented with a small dime (heads side on a white background) to signify the end of a low reward maze.

Procedure: To control for the effect of outcome location on the retroactive effect of reward effect, we conducted Experiment 3, which replicated the 24-hour condition of Experiment 1, except that we replaced the no-reward maze over condition with a low reward condition. This way, the maze outcome—either a gold coin (still worth a bonus $1) or a dime (worth a bonus $0.10)—was always presented in a maze square. The instructions were the same, except that the instructions for the Phase 1 maze exploration (incidental encoding) task were modified: "In each maze, either a gold dollar coin or a dime is hidden. You will navigate through each maze to find the hidden coin. You will be paid a $1 bonus for every gold coin that you find. You will be paid 10 cents for every dime that you find."

**Experiment 4**. We conducted Experiment 4 to test whether the reward proximity effect and the reward proximity by rest duration effect were supported by strategic rehearsal during the post-encoding rest breaks. Experiment 4 was identical to the 24-hour condition of Experiment 1 and to Experiment 2, except that during the maze encoding task, we replaced the rest breaks with one of three distractor tasks designed to interfere with strategic rehearsal at different levels of cognitive difficulty: a target detection task, a spatial navigation task, and a working memory task. In Experiment 4, participants had a low rate of repeated objects (2.25 ± 2.62% of all trials on average) or "Too Late" missed trials (4.70 ± 5.03% of all trials on average).

Participants: For Experiment 4, we recruited 88 participants. We excluded five participants due to self-reported psychiatric diagnoses and/or psychoactive medication, four participants due to technical errors, and two participants for previously participating in a similar task, resulting in 77 participants (43 female, mean age (SD): 23.9 ± 5.0 years). Of these, we assigned 27 participants to the target detection task, 27 participants to the spatial navigation condition, and 23 participants to the working memory condition. Due to technical errors, only 75 participants completed the Phase 3: surprise spatial location memory test.

Procedure: Experiment 4 was identical to the 24-hour condition of Experiment 1, except that during the 15, 20 or 25-s intervals following each maze, participants performed one of three distractor tasks with varying levels of cognitive load (described below) designed to prohibit strategic rehearsal during the breaks. In the practice session, participants practiced the distractor task following each of the two practice mazes. To help participants differentiate between the maze encoding task and the distractor task, at the onset of the distractor task, the screen's background colour changed from the light grey used during the maze exploration to a darker grey.

Target detection task: Participants were instructed to press the up arrow key each time a target (a black and white bulls eye image) appeared. Throughout the post-maze interval, a blank maze grid was presented. Intermittently, a stimulus—either the target (25% of trials) or a lure (a different coloured grey square, 75% of trials)—appeared in a randomly determined square within the maze. The stimulus was presented for 2 s, and the participant needed to respond to the target while the stimulus remained on the screen. If the participant did not respond within the allotted time or if an inaccurate response was made, a warning appeared at the top

of the screen. A jittered interval (mean: 2.3 s) with a blank maze grid separated trials.

Spatial navigation task: Participants were instructed that a target would appear in a blank maze and that they should use the up, down, left and right arrow keys to navigate to the target (again, a black and white bulls eye image). Throughout the post-maze interval, a blank maze was presented, in which a target would appear in a randomly chosen square within the maze. The participants' starting location was indicated with a white frame surrounding another randomly chosen square. Participants had 2 s to make a navigational choice. If participants did not make a choice within the response window, a warning appeared at the top of the screen. The next location square would turn white for 0.5–1.5 s (mean: 1.25 s). The next choice period was signalled by the square returning to the grey background colour. Participants navigated to a series of targets for the duration of the inter-maze interval.

Working memory task: Participants were instructed that at the beginning of the post-maze interval, a target would appear and that they should press the up arrow any time an identical probe appeared during the series of probe stimuli that followed. The target was a blank maze with four randomly chosen squares appearing in colour (any combination of red, orange, yellow, green, blue or purple, randomly chosen). Next, a series of probe stimuli appeared for 1.75 s: targets (20%), location lures (the same locations as the target, but different colours, 10%), colour lures (the same colours as the target, but different locations, 10%), and other lures (50%). A novel target was used after each maze.

**Analyses**. We conducted all pre-processing in Matlab and analyses in R. Since our goal was to relate memory for a single item based on trial-by-trial variables such as reward and proximity, the majority of the recognition memory analyses we report in the manuscript include only responses for old objects, as the lures were not associated with the necessary trial-by-trial variables. For the analyses of recognition memory (Phase 2), we used a multi-level logistic regression model to predict hits (i.e. an old object correctly identified as old) in R (glmer, in the lmer4 package).

For the analyses of spatial location memory (Phase 3), we operationalized spatial memory by measuring the number of steps between the encoded location and the remembered location, and then, because the maximum possible error varied as a function of the object's original encoding location (i.e. it is possible to have an error of eight steps for an object encoded in the corner, but only four steps away from an object encoded in the centre of the maze), we scaled each error by the maximum possible error for that encoded location. Then we subtracted this score from 1 so that each trial was given a location memory score ranging from 0 (the remembered location was as far away as possible from the encoded location) to 1 (the encoded location was remembered correctly). Additionally, we also ran a parallel set of analyses that used the raw spatial location memory error. For the spatial location memory analyses, we only included trials for which the participant correctly identified the object as an old object (i.e. a hit) during the recognition memory test (Phase 2).

We operationalized proximity as the number of sequential steps between the object and the outcome (reward or no reward), such that the object seen immediately before the maze end was one step away, the object before that was two steps away, and so forth up to fifteen steps away. We operationalized spatial proximity as the number of maze steps, in any direction, between an object and the end square (minimum one step, maximum eight steps). Any object seen more than once during maze exploration due to a participant retracing his or her step was removed from all analyses.

For the models, reward was effect-coded (1 for reward, −1 for no reward or low reward), proximity was mean-centred and scaled by subject, spatial proximity was mean-centred and scaled by subject, and rest duration was mean-centred and scaled by subject. In Experiment 1, the delay condition was effect-coded (1 for 24 hour, −1 for 15 minute), and in Experiment 4, the distractor condition was coded as a factor. For the spatial location memory test, we additionally included Experiment, coded as a factor, as a predictor.

In the main reward × proximity interactions, we fit separate intercept, reward, proximity, and reward × proximity interaction effects for each subject. For the models that included rest duration, we additionally added random effects for rest, the rest × reward interaction, the rest × proximity interaction, and the rest × reward × proximity interaction. We did not model correlations between the random effects across subjects. Models that did not converge were rerun using the bobyqa optimizer increased to 1,000,000 iterations.

We estimated the confidence intervals using the confint.merMod function and the p-values using the bootMer function (both from the lmer4 package) run with 2500 iterations. All p-values are two-tailed. To compare the model fits, we used likelihood ratio tests implemented with the anova function.

To ensure that our results were not due to modelling the data using a mixed effect model, we additionally fit a separate reward × proximity model to each participant's data so that subject is treated as a fixed effect. We then used a linear regression model to fit the reward × proximity interaction effect by group (1 for 24 hour, −1 for 15 minute) across subjects.

**Data availability**
The data sets collected for these experiments and data analysis code are available in the Open Science Framework repository, https://doi.org/10.17605/OSF.IO/GZ9XE.

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

## Acknowledgements

This research was supported by the National Institute of Health (5R01DA038891 to D. S.), the McKnight Foundation (MCKNGT CU16-0460 to D.S.) and the National Science Foundation (Career Award BCS-0955494 to D.S. and Graduate Research Fellowship DGE-1144155 to E.K.B.). We thank Mariam Aly, Niall Bolger, Nathaniel Daw, Katherine Duncan, Raphael Gerraty, Greg Jensen, Dave Krantz, and Travis Riddle for insightful conversations and advice; Sadie Bennett, Adam Litt, Lucy Owen, Anuya Patil, Blair Vail, and Camilla van Geen for their help with data collection; and Eileen Hartnett and Camilla van Geen for comments on an earlier draft.

## Author contributions

E.K.B., G.E.W and D.S designed the experiment. G.E.W and E.K.B. programmed the experiment. E.K.B. supervised data collection. E.K.B. performed the data analysis. E.K.B., G.E.W. and D.S. wrote the manuscript.

## Additional information

**Competing interests:** The authors declare no competing interests.

