## [Peer Review File · Nature Communications]

Reviewers' comments:

Reviewer #1 (Remarks to the Author):

The authors introduced many theories of how retroactive memory enhancement can occur via increased replay. However, I thought manuscript was a bit misleading with respect to the present study because replay/reactivation was not directly examined or tested. Though one might argue that increased time between mazes allows more time for replay, there can be other processes that could be occurring that lead to the pattern of results that were found. This limitation of the present finding needs to be acknowledged in both the framing and the Discussion. As one example, there is no evidence upon which the authors can make this conclusion, "these results are consistent with neurophysiological findings in rodents demonstrating that replay of experiences immediately following encoding and during sleep is important for subsequent choice behavior."

While the design was fairly straightforward, a number of concerns should be addressed. In the manuscript, a lot of emphasis is placed on their ability to index memory strengthening due to replay. They specifically refer to place cells and refer to their task as a maze paradigm that participants have to navigate. Despite the active role participants have in moving themselves through the maze/grid, the sequences are not spatially dependent. In fact, spatial position plays no role in the task. All that separates the end of a "maze" from the next trial is presentation of a "maze over" display. Displaying sequences of objects on the screen that are either rewarded or not would have been equivalent. While I do agree that offline processing is facilitating memory updating, I believe the explicit replay/spatial aspects of their manuscript should be downplayed. Perhaps there were location-dependent tests of memory that are not reported that might allow one to distinguish whether the reward-driven memory enhancement is due to spatial or temporal proximity to reward?

I had minor concerns about the lack of an "end" marker in the no-reward condition. I understand that participants were aware the maze was over because of the "MAZE OVER" notification, but this notification is different from the coin cue, which was integrated into the maze itself and implicitly signaled the end of the maze. Though unlikely, perhaps the effects are driven simply by proximity to the "end" marker. A more appropriate control would be to have people run through mazes for which half ended in a gold coin (indicating reward), and the other half ended with a lump of coal or some other consistent object marker that indicates the end of a no-reward maze trial.

With respect to the immediate time interval manipulation, even the shortest delay (15 seconds) seems like sufficient time for replay (see work on replay by Staresina that indexed reactivation during short intervals of rest). Additional justification is therefore needed as to why the 3 different time delays (15s, 20s, 25s) were chosen, and/or a more compelling explanation as to why the reported effects emerge only with the 25s break interval between mazes.

Why did the surprise recognition memory test consist of only 96 lures (vs. 192 old items)?

Confidence judgments were collected as part of the design, but there was no explanation as to why these measures were collected, nor was there any reporting of these results (ROC curves, etc).

I thought Figure 2 was a bit oversimplified and not extremely informative. From the supplemental figures, it appears there is tremendous individual variability in the data, and it would be important to incorporate the distribution of data into the summary figure.

In figure S2c, it appears that more than half of the participants fail to show the pattern of behavior that is reported in the results, and at least 3 participants show behavior that looks almost the opposite

of the reported finding—for instance, subject 21 shows a pretty strong primacy effect in the reward condition. What does this say about the hypotheses/predictions suggested in the paper? In the “effects of maze length on memory” section of the supplement, it’s unclear whether they’re testing the interactions on all trials or only hit trials. I’d be interested in what the interactions would be including hit rate.

As a behavioral paper, the number of participants in each group (N = 21 in the 15 minute condition and N = 23 in the 24 hour condition) is low. This might be due to the fact that this is perhaps the behavioral data for an imaging study that they’re collecting. But as a stand alone manuscript, the Ns seem low. This is especially concerning given the individual differences mentioned above.

Reviewer #2 (Remarks to the Author):

In this paper, Kendall Braun et al. investigate how unexpected rewards retroactively influence the formation of memories. Participants were exposed to incidental objects while navigating around a maze. Occasionally, a spatial trajectory was rewarded. Consistent with recent findings in the animal literature, the authors find that reward influences memory strength along a gradient, with a stronger retention of objects experienced just before a reward was experienced, and a weaker retention of objects experienced a longer time ago. This retention depends both on overnight consolidation and the duration of a post-encoding rest period.

This study addresses a timely question using a clever experimental design and the findings will be of interest to other researchers in the community. It is also nice to see that the authors replicate their findings multiple times within the same study. However, I have three major concerns:

1) The authors base the interpretation of their results on a negative effect of proximity on memory in the rewarded mazes. However, looking across all distances, there doesn’t seem to be any benefit of reward for memory retention, at least in the 24h condition. Furthermore, in many of the conditions there is not just a negative effect of distance in rewarded mazes, but also a positive effect of distance in the unrewarded mazes (particularly evident in Figure 4). While the positive slope is probably not significant in most conditions, it seems that objects that are further away from the final location of a trajectory are generally better remembered if no reward is encountered compared to a rewarded situation. In other words, it seems that the interpretation that distal events are forgotten if reward is encountered may be more appropriate than the interpretation that proximal events are remembered in rewarded mazes. I am not sure how this can be reconciled with the idea that reward-induced reverse replay strengthens the memory of proximal events. The authors seem to ignore this intriguing observation. Do they have an explanation for this?

On a related note, does the hit rate suggest that participants perform at chance (at least in the 24h plot)? If so, I am not sure how meaningful the scaling by proximity is. Again, it would be good if the authors could comment on this.

2) In their plots, the authors only display the results of the model fit to their data, rather than the data itself. It would be much more informative to see the raw data plotted (at least in addition to the model fits), also because this would help assess whether the linear model is a good fit at all.

3) Can the effect of reward and proximity on memory be explained by other confounding factors? For example:

a. The post-object jitter length, which varies considerably between objects (this is unlikely to vary with distance though, if randomised appropriately)

b. The position of an object on the maze. Participants may have a tendency to explore particular locations, either because they are more salient (e.g. central locations) or because they were previously rewarded. For example, locations in the centre of the maze may be a lot more salient than locations in the surrounding areas. Because of their salience objects at these locations may be easier to remember. But because participants keep navigating to these locations, objects at these locations will also occur closer in time to the reward. The relationship between memory and distance to reward may therefore be incidental.

c. The number of objects experienced in any given position in space, either before encountering the object of interest or in total (i.e. the number of times a given location has been visited before). This is again related to the reasoning outlined in b).

d. The number of times a reward was encountered at a given object's location in space (either before an object of interest, or in total). Again related to the reasoning outlined in b).

Minor comments:

- The stimuli are arranged on a spatial maze structure. Temporal succession is therefore not the only conceivable distance measure between objects and rewards. Instead, spatial proximity on the maze may also influence memory. It would be very interesting (albeit not necessary) if the authors found such effects in their data as such a result would nicely add to a range of recent studies investigating cognitive maps in the hippocampus.

- With its strong focus on hippocampal coding mechanisms and replay, the introduction reads as if this is a study investigating replay in humans. Maybe make it clearer that study does not directly look for a neural signature of replay, but instead tests behavioural predictions of the phenomenon. The introduction could in general be a bit more concise.

- How were participants instructed? Did they receive any information as to where a reward would be found? Could they have been under the impression that (a) rewards are distributed randomly and it is therefore advisable to keep searching for rewards in different locations on different trials or that (b) rewards are more likely to be found in certain parts of the maze?

Reviewer #3 (Remarks to the Author):

Braun and colleagues report a series of behavioral studies that look at the retroactive influence of reward on memory. In the paradigm they used, subjects explored a "maze" (or grid), uncovering objects at each location until they either discovered a reward (a coin) or the trial ended. Of critical interest was whether finding the reward (coin) influenced memory for the preceding objects in a temporally-graded manner, with preferential memory enhancement for the items closest (in time) to the reward. Indeed, across several studies (essentially 5 independent variants), they find a highly consistent pattern of results where memory for the objects varies according to reward AND proximity to the reward (i.e., there is an interaction). This interaction is driven by a negative slope relating position to memory in the reward condition (worse memory for objects further from the reward) and a generally positive slope for the non-reward condition (relatively better memory for objects further away from the reward). However, this pattern of results was only observed after a 24-hour delay and it also increased as a function of the amount of time that followed each reward cue (i.e., the inter-trial interval).

The results are very intriguing and potentially establish some novel, important results. The way the

paper is framed is compelling and nicely motivates the ideas. Moreover, the multiple replications make a strong case that the effect is "real." The sensitivity of the effect to delay and to the post-reward time interval are interesting results, as well. That said, I am less convinced that the actual mechanism driving the results has been worked out. The results are intriguing enough that this may not necessarily be a fatal flaw, but I think there are some important points that need to be addressed/considered. Additionally, the paper could benefit from some editing as I think the presentation of results is a bit confusing.

Major Comments

1. Lack of reward enhancement. To me, the "elephant in the room" is that there is essentially no evidence that reward had an overall positive influence on memory. If anything, the only hint of an effect was in the 15 minute delay condition. But in the 24 hour delay conditions, there is no evidence for a main effect of reward. Thus, claims of "enhancement of memory by reward" (as appears in the title and elsewhere in the paper) are misleading. Likewise, there are statements such as "reward retroactively and selectively enhanced memory." Clearly, there are differences between the reward condition and the no reward condition, but I am not sure it can be called enhancement. Even if specifically considering the single item just before the reward cue (or end of the trial), I don't believe the authors actually report whether memory for this individual item is significantly enhanced. I am assuming it is, but even if it is, this enhancement seems to come at the expense of memory for other items. Thus, it is more like a reprioritization that is induced by reward. Whatever enhancement there might be, there are offsetting costs. The lack of an overall effect of reward was not considered in the Discussion, but I think this point warrants explicit discussion.

1a. Another example related to the reward enhancement claims: In Fig1a, it states "In the 24-hour condition, reward retroactively modulated memory, such that participants were more likely to remember objects that were more proximal to the reward." But the statistic that is reported here is actually the interaction between reward and proximity. To say that reward modulated memory such that participants were more likely to remember objects that were proximal to the reward, they should report whether the slope for the reward condition (alone) was significant. They don't seem to report this. While they do show the mean beta for the reward condition in the inset, the error bars here are from the interaction, so it's not possible to assess whether the beta is significantly negative for the reward condition alone.

2. Is the reward effect actually a reward effect? Again, there is clearly a difference between the reward and no reward conditions. But, I am wondering whether this effect is due to reward, per se. I realize this may sound like a subtle point, but the reward and no reward conditions are not perfectly matched in that the reward trials involved presentation of the coin at a location ON THE GRID (and necessarily spatially adjacent to the immediately preceding object), whereas the no reward trials involved presentation of a message ABOVE THE GRID (and this message was of course not spatially adjacent to the immediately preceding object). This had me wondering whether the "MAZE OVER" screen might have functioned to draw subjects' attention away from the grid and disrupted memory for the preceding items. As I note above, the reward effect appears to be driven both by the negative slope for the reward condition and a positive slope for the no reward condition. This raises the question of why there was a positive slope for the no reward condition. Perhaps this simply reflects buildup of interference across the list, but it could also reflect a disruption from the "MAZE OVER" screen. The most obvious solution to this issue would be to better match the reward and no reward trials such that for the no reward trials a "worthless" coin appeared in precisely the same spot that the reward coin would appear. But, with the current procedure, I am not entirely convinced that the reward effect is related to reward, per se. The alternative account would be something about interference/distraction. Of course, even with this alternative account, there are still the interesting

facts that the influence emerges over time (24 hour delay) and that it is sensitive to the inter-trial interval.

3. The fact that there is a positive slope for the no reward condition also seems to go against the idea in the Introduction that replay occurs in reverse order after each event. If this were true, then we should expect a negative slope for BOTH the reward and no reward conditions but that the slope would simply be steeper and/or shifted up for the reward condition.

4. All of the effects relating temporal position to memory are based on linear relationships. Are the effects actually linear? Relatedly, I am wondering how much of the effect is driven by memory for the single item prior to the reward (or "MAZE OVER" cue)? It would be helpful to see the data plotted as a function of number of steps from the end point (i.e., $n-1$, $n-2$, etc.). As is, we only see the fitted lines, not any of the actual mean data.

5. The fact that the reward effect requires 25 seconds of inter-trial interval is interesting, but I also find these results surprising. The argument is that events are replayed in the delay condition, but isn't 15 seconds (the shortest interval) enough time to at least replay the last couple of objects? If it were a difference of 1s vs. 10s, that would be more intuitively reasonable to me, but 15 seconds seems like an awfully long time (i.e., enough time for replay). I realize the data are the data, but I am curious if the authors have any intuitions about the timing and/or how they settled on these numbers.

6. Related to the above point: if the putative replay mechanism is not disrupted by the distractor tasks (target detection, working memory, navigation), then why would/should it be disrupted by the onset of the next trial? In other words, it almost seems like the replay should occur no matter what, meaning that the inter-trial interval length would be irrelevant.

7. The way experiments 1 and 2 are split between the main text and the supplement is confusing. The main text refers you to the supplementary figures for the results of Experiment 2, but then you need to go to the methods to figure out what Experiment 2 actually involved. Since experiment 2 was a "straight replication" why not just include the replication statistic in the main Results section? Also having the figures for Experiments 1 and 2 split across the main text and Supplement makes it more of a hassle to compare.

Minor Comments

8. Statistics are not always reported in the main text and in some cases, it is hard to follow which experiment the data are coming from. For example, for the section describing the reward proximity effect as a function of rest duration, I initially thought this was data from a new experiment. The figure caption does say "experiment 1" but the main text does not list an experiment number or otherwise make it clear that the data are from experiment 1.

9. In Supplementary Figure 3b, are the betas shown in the inset correct? They appear to be swapped or otherwise incorrect in that the mean beta for the reward condition is shown as negative but the slope of the line in the main part of the figure is positive.

10. Please elaborate on why subjects were excluded for "non-compliance."

11. On p. 9 the text says "These results suggest that the short time window following encoding is a prerequisite for the subsequent differential consolidation of these memory traces." I found this sentence confusing. At first, I thought the point of the sentence was to emphasize the short time

window CONDITION (i.e., the 15 second condition) but I believe instead the authors simply mean that the delay period following each maze was important.

Reviewer #1 (Remarks to the Author):

The authors introduced many theories of how retroactive memory enhancement can occur via increased replay. However, I thought manuscript was a bit misleading with respect to the present study because replay/reactivation was not directly examined or tested. Though one might argue that increased time between mazes allows more time for replay, there can be other processes that could be occurring that lead to the pattern of results that were found. This limitation of the present finding needs to be acknowledged in both the framing and the Discussion. As one example, there is no evidence upon which the authors can make this conclusion, “these results are consistent with neurophysiological findings in rodents demonstrating that replay of experiences immediately following encoding and during sleep is important for subsequent choice behavior.”

We thank the reviewer for raising this important point. Although the experimental question and experimental design were motivated by our desire to link findings from rodent electrophysiology research to human episodic memory, we agree that with this series of behavioral experiments we can only indirectly test the effects of post-encoding mechanisms on memory. We certainly do not wish to inadvertently mislead our readers and have revised the introduction to emphasize the relevant behavioral literature (see introduction p. 3). Further, in the revised manuscript, we now describe the relevant rodent electrophysiology research in the discussion (see p. 10), where we are careful to point out the speculative nature of these links between this research and our own results. We also modified the abstract accordingly.

Additionally, we have clarified our language throughout the manuscript to ensure that we are communicating our findings precisely (see introduction p. 3 and discussion p. 10). We have removed the sentence “these results are consistent with neuropsychological findings in rodents demonstrating that replay of experiences immediately following encoding and during sleep is important for subsequent choice behavior” (see discussion p. 10).

While the design was fairly straightforward, a number of concerns should be addressed.

In the manuscript, a lot of emphasis is placed on their ability to index memory strengthening due to replay. They specifically refer to place cells and refer to their task as a maze paradigm that participants have to navigate. Despite the active role participants have in moving themselves through the maze/grid, the sequences are not spatially dependent. In fact, spatial position plays no role in the task. All that separates the end of a “maze” from the next trial is presentation of a “maze over” display. Displaying sequences of objects on the screen that are either rewarded or not would have been equivalent. While I do agree that offline processing is facilitating memory updating, I believe the explicit replay/spatial aspects of their manuscript should be downplayed. Perhaps there were location-dependent tests of memory that are not reported that might allow one to distinguish whether the reward-driven memory enhancement is due to spatial or temporal proximity to reward?

Thank you for raising the important question of the role of space in our paradigm. Indeed, while the analyses in our original manuscript focused only on sequential proximity, a central feature of our experimental design is that objects are encoded during exploration in a spatial-temporal context, and our original manuscript did not adequately leverage this richness. We have addressed the reviewer’s comment in three ways.

First, we added new analyses of a spatial location memory test, which was collected after the recognition memory test in each experiment. As described in the revised manuscript (see results p. 9 and online method p. 17), to test object location memory, an old object was randomly placed in the maze and the participant was instructed to move the object back to the square where the object was originally encoded. We measured the number of steps between the encoded location and the remembered location. Since the maximum possible error varied as a function of the object's original encoding location (i.e. it is possible to have an error of eight steps for an object encoded in the corner, but only four steps away from an object encoded in the center of the maze), to compute an accuracy measure, we scaled each error by the maximum possible error for that particular encoding location. Then we subtracted this score from 1, so that each trial was given a location memory score ranging from 0 (the remembered location was as far away as possible from the original location) to 1 (the location was remembered correctly).

For these analyses, we combined all of the data from all four experiments when memory was tested after 24-hours. First, we found that participants' spatial location memory performance was above chance (mean spatial memory (se) = 0.54 ± 0.00030 , $t(145) = 13.36$, $p < 0.001$). Second, in an analysis that paralleled the results from the recognition memory test, we found that reward retroactively modulated spatial memory for sequentially proximal objects (data from all Experiments 1, 2, 3 and 4, combined: Reward X Proximity: $\beta = -0.0062$, $SE = 0.0022$, $t = -2.83$, $CI_{95} = [-0.010, -0.0018]$, $p = 0.0032$). Third, we found a main effect of reward (data from all four experiments combined: $\beta = 0.0054$, $SE = 0.0022$, $t = 2.47$, $CI_{95} = [0.0011, 0.0095]$, $p = 0.019$), such that spatial location memory for the rewarded mazes was significantly better than the non-rewarded mazes. We have incorporated these results into the manuscript (see results p. 9, figure 5, online method p.17, supplement p. 24, p. 34).

Further, these results were not simply a by-product of the chance-corrected scoring method: we found the same pattern of results when we conduct the analysis using the uncorrected error score (i.e. the number of steps error between the encoded location and the remembered location). Note that since this measures error in memory performance, better memory is indicated with a lower error (results from Experiments 1-4 combined: reward x proximity: $\beta = 0.037$, $SE = 0.014$, $t = 2.57$, $CI_{95} = [0.0094, 0.066]$, $p = 0.0056$; reward: $\beta = -0.040$, $SE = 0.014$, $t = -2.80$, $CI_{95} = [-0.068, -0.013]$, $p = 0.0032$; see supplemental results p. 34). We acknowledge that these effects are small, but in combination with the parallel effects in the recognition memory test, these results suggest that spatial memory for a cognitive map is retroactively modulated by reward.

Second, we wanted to explore how rewards retroactively modulated recognition memory for spatially proximal objects. When we tested to see if there was an interaction between reward and *spatially* proximal objects, we found that rewards retroactively modulated memory for spatially proximal objects (or a trend in the same direction) in all four experiments (reward x spatial proximity; Experiment 1 – 24-hour condition: $\beta = -0.086$, $SE = 0.032$, $z = -2.67$, $CI_{95} = [-0.15, -0.022]$, $p = 0.011$; Experiment 2: $\beta = -0.067$, $SE = 0.035$, $z = -1.92$, $CI_{95} = [-0.13, -0.00097]$, $p = 0.044$; Experiment 3 (new to the revision): $\beta = -0.038$, $SE = 0.027$, $z = -1.41$, $CI_{95} = [-0.089, 0.014]$, $p = 0.17$; Experiment 4 (Experiment 3 in the original manuscript) – target condition: $\beta = -0.075$, $SE = 0.029$, $z = -2.59$, $CI_{95} = [-0.19, -0.063]$, $p = 0.011$; navigation condition: $\beta = -0.067$, $SE = 0.030$, $z = -2.26$, $CI_{95} = [-0.13, -0.02]$, $p = 0.011$; working memory condition: $\beta = -0.056$, $SE = 0.032$, $z = -1.76$, $CI_{95} = [-0.12, 0.0070]$, $p = 0.082$). One caveat to this analysis is that in our task participants navigated through the mazes freely, and consequently, sequential proximity and spatial proximity

are correlated (spatial proximity ~ sequential proximity: Experiment 1 – 24 hour condition: $\beta = 0.78$, $SE = 0.035$, $t = 22.44$, $CI_{95} = [0.71, 0.85]$, $p < 0.0004$, Experiment 2: $\beta = 0.78$, $SE = 0.029$, $t = 26.87$, $CI_{95} = [0.72, 0.83]$, $p < 0.0004$; Experiment 3: $\beta = 0.79$, $SE = 0.023$, $t = 34.24$, $CI_{95} = [0.75, 0.84]$, $p < 0.0004$, Experiment 4 – target condition: $\beta = 0.72$, $SE = 0.035$, $t = 20.65$, $CI_{95} = [0.65, 0.79]$, $p < 0.0004$; navigation condition: $\beta = 0.74$, $SE = 0.036$, $t = 20.74$, $CI_{95} = [0.67, 0.81]$, $p < 0.0004$; working memory condition: $\beta = 0.80$, $SE = 0.034$, $t = 23.30$, $CI_{95} = [0.73, 0.87]$, $p < 0.0004$). When we controlled for the reward by sequential proximity interaction, we did not detect evidence that reward retroactively modulates memory by spatial proximity (reward x spatial proximity; Experiment 1: $\beta = -0.039$, $SE = 0.038$, $z = -1.04$, $CI_{95} = [-0.11, 0.034]$, $p = 0.30$; Experiment 2: $\beta = -0.0030$, $SE = 0.040$, $z = -0.076$, $CI_{95} = [-0.080, 0.077]$, $p = 0.94$; Experiment 3: $\beta = 0.038$, $SE = 0.032$, $z = 1.18$, $CI_{95} = [-0.024, 0.10]$, $p = 0.24$; Experiment 4 – target condition: $\beta = -0.0094$, $SE = 0.033$, $z = -0.28$, $CI_{95} = [-0.073, 0.055]$, $p = 0.78$; navigation condition: $\beta = 0.0079$, $SE = 0.034$, $z = 0.23$, $CI_{95} = [-0.057, 0.077]$, $p = 0.84$; working memory condition: $\beta = 0.015$, $SE = 0.037$, $z = 0.40$, $CI_{95} = [-0.058, 0.085]$, $p = 0.66$). However, in each of these analyses, we still detected evidence of the reward proximity effect (reward x sequential proximity; Experiment 1: $\beta = -0.086$, $SE = 0.038$, $z = -2.28$, $CI_{95} = [-0.16, -0.011]$, $p = 0.024$; Experiment 2: $\beta = -0.12$, $SE = 0.040$, $z = -3.09$, $CI_{95} = [-0.20, -0.040]$, $p = 0.0016$; Experiment 3: $\beta = -0.14$, $SE = 0.032$, $z = -4.43$, $CI_{95} = [-0.20, -0.079]$, $p < 0.0004$; Experiment 4 – target condition: $\beta = -0.13$, $SE = 0.033$, $z = -4.03$, $CI_{95} = [-0.20, -0.69]$, $p < 0.0004$; navigation condition: $\beta = -0.15$, $SE = 0.034$, $z = -4.44$, $CI_{95} = [-0.22, -0.085]$, $p < 0.0004$; working memory condition: $\beta = -0.13$, $SE = 0.038$, $z = -3.44$, $CI_{95} = [-0.21, -0.056]$, $p < 0.0004$). Together, these results suggest that sequential proximity explains the data better for than spatial proximity, as would be predicted by models of hippocampal replay. We now include these additional analyses in the manuscript (see supplement p. 27).

Third, as described above, we have downplayed our discussion of place cells in the introduction, emphasizing the relevant psychological literature, and instead speculate about the role of place cells as a mechanism for the reward proximity effect in the discussion (see introduction p. 3, discussion p. 11).

I had minor concerns about the lack of an “end” marker in the no-reward condition. I understand that participants were aware the maze was over because of the “MAZE OVER” notification, but this notification is different from the coin cue, which was integrated into the maze itself and implicitly signaled the end of the maze. Though unlikely, perhaps the effects are driven simply by proximity to the “end” marker. A more appropriate control would be to have people run through mazes for which half ended in a gold coin (indicating reward), and the other half ended with a lump of coal or some other consistent object marker that indicates the end of a no-reward maze trial.

Thank you for identifying this concern. To address it, we ran a new experiment (“Experiment 3”, see results p. 6, p. 8, Figure S1c, Figure S3b,d, online method p. 19). We opted to isolate the effect of reward as conservatively as possible by comparing low vs. high rewards, removing the no reward (“maze over”) condition altogether. In this new experiment, participants navigated through a series of mazes that ended either in a dime (worth \$0.10 bonus) or the same gold coin reward used previously (worth \$1 bonus), so that the maze outcome always appeared as a coin within the maze.

The results of Experiment 3 replicated the reward proximity finding described in the original manuscript: participants had better memory for the objects leading up to the gold coin relative to objects leading up to the dime (reward x proximity: $\beta = -0.12$, $SE = 0.027$, $z = 4.48$, $CI_{95} = [-0.17, -$

0.071], $p < 0.0004$). We also found, again, that the duration of the post-encoding rest break interacted with the reward proximity effect, such that the reward proximity effect was stronger when the post-encoding rest break was longer (reward x proximity x rest duration: $\beta = -0.018$, $SE = 0.035$, $z = -5.06$, $CI_{95} = [-0.25, -0.11]$, $p < 0.0004$) and that the model including rest duration explained the data significantly better than the simpler model omitting rest duration ($\chi^2(8) = 38.90$, $p = 0.0000051$), replicating the effects reported originally. Together, these results suggest that the pattern of results reported in our original manuscript cannot be explained by differences related to the maze outcome being presented in the maze, but instead are due to the value of the reward.

With respect to the immediate time interval manipulation, even the shortest delay (15 seconds) seems like sufficient time for replay (see work on replay by Staresina that indexed reactivation during short intervals of rest). Additional justification is therefore needed as to why the 3 different time delays (15s, 20s, 25s) were chosen, and/or a more compelling explanation as to why the reported effects emerge only with the 25s break interval between mazes.

Thank you for giving us an opportunity to explain our motivation for choosing the duration of the rest intervals. When we designed this experiment, we were motivated by evidence emerging from the rodent literature that linked hippocampal replay to memory and endeavored to develop a human experiment that paralleled these experiments in a reasonably close way. We initially chose a rest duration that loosely approximated the length of an inter-trial interval in such experiments, and then, as we plan to run an fMRI experiment to measure brain activity during these intervals, we increased the time windows to acquire more fMRI data during this critical time period. In piloting, we found that these time windows effectively modulated memory, and consequently, we have not changed them in the ensuing experiments. We have added an explanation for how we chose these time durations in the discussion (see discussion p. 11).

In the discussion (see p. 11), we also now explicitly address the point the reviewer raises, which is that it is not entirely clear from the replay data in rodents why a longer interval would be needed. Because hippocampal replay in rodents and post-encoding reactivation in human neuroimaging persists for an extended period of time following encoding, we speculate that the amount of replay, as opposed to a single instance of replay, may be important.

Why did the surprise recognition memory test consist of only 96 lures (vs. 192 old items)?

It is not uncommon for experiments with subsequent recognition memory tests to use fewer number of lures than old stimuli for practical reasons [for example see Gruber, Ritchey, Wang, Doss, & Ranganath, 2016 (320 old items, 120 lures); Chowdhury, Guitart-Masip, Bunzeck, Dolan, Duzel, 2012 (60 old items, 30 lures); Ritchey, Montchal, Yonelinas, & Ranganath, 2015 (168 old items, 84 lures); Wimmer, Braun, Daw, & Shohamy, 2014 (200 old items, 100 lures); and Dimsdale-Zucker, Ritchey, Ekstrom, Yonelinas, & Ranganath (200 old items, 52 lures)].

In our experiments, we chose to use fewer lures to shorten the duration of the experiment and avoid unnecessary fatigue. In Experiment 1, the 15-minute condition lasted almost 150 minutes, a long time to remain focused on cognitively demanding tasks. In subsequent experiments, we maintained the same number of stimuli (old objects and lures) for methodological consistency. We revised the online method to include our rationale for this decision (see p. 17).

Confidence judgments were collected as part of the design, but there was no explanation as to why these measures were collected, nor was there any reporting of these results (ROC curves, etc).

Thank you for raising this issue. We collected confidence ratings for the memory judgments, as this is standard practice for recognition memory tasks (for example, see Wimmer, Braun, & Shohamy, 2014; Foerde, Braun, & Shohamy, 2013; Pezdek, 1978; Dunsmoor, Murty, Davachi, & Phelps, 2015; Davachi, Maril, & Wagner, 2001), and we now clarify our motivation for collecting these responses in the online method (see p. 17).

We looked into the possibility of conducting ROC analyses, but unfortunately, we do not have enough data to conduct this type of analysis. For ROC analyses, it is recommended that for each participant there be at least 60 old trials and 60 new trials for each condition (Yonelinas & Parks, 2007). Since our analysis of interest predicted memory based on reward (i.e. reward vs. no reward) and proximity, even if we simply divided the proximity measure into two levels (e.g. proximal and distal), then we would require at least 480 trials per participant; however, we only collected 288 memory responses.

I thought Figure 2 was a bit oversimplified and not extremely informative. From the supplemental figures, it appears there is tremendous individual variability in the data, and it would be important to incorporate the distribution of data into the summary figure.

We appreciate the reviewer's feedback regarding our figures, as we want to convey as much information about the data as possible. Although the reward x proximity effect is very reliable across all six datasets tested after 24-hours, you appropriately identify that there is variability in each subject's predicted lines, as would be expected from any memory test involving incidentally encoded stimuli. Further, we would expect the subject-level prediction plots to be noisy, as each participant has only one data point for reward and no reward for at the 13-steps, 14-step, and 15-step proximity.

Therefore, to help our readers visualize this variability, in addition to plotting interaction term using the beta plots as we did in the original manuscript, we now include the interaction term plotted by proximity (i.e. the difference between the predicted reward line and the predicted no (low) reward line) at the group level and also for each subject (see Figure 2). This approach allows readers to visualize the difference between the reward and no reward prediction lines (i.e. the interaction), while simultaneously illustrating the subject-level variability.

Additionally, in the revised manuscript, we now include a figure plotting mean memory as a function of reward and proximity (see Figure S1e).

In figure S2c, it appears that more than half of the participants fail to show the pattern of behavior that is reported in the results, and at least 3 participants show behavior that looks almost the opposite of the reported finding—for instance, subject 21 shows a pretty strong primacy effect in the reward condition. What does this say about the hypotheses/predictions suggested in the paper?

Thank you for raising the question of individual differences in the reward by proximity effect.

The comment about individual variability prompted us to consider the overall pattern of the effects and to add new summary statistics across all the datasets. In particular, we want to

emphasize that despite this variability, the pattern of effects at the group level are very reliable across six datasets: We found a reward by proximity effect in the same direction for 78.3% of participants in Experiment 1 – 24-hour condition, 85.7% of participants in Experiment 2, 81.3% of participants in Experiment 3, and 96.3% of participants in Experiment 4 – target detection condition, 88.9% of participants in Experiment 4 – navigation condition, and 73.9 % of participants in Experiment 4 – working memory condition. We have added these statistics to the supplement (see p. 26).

In the “effects of maze length on memory” section of the supplement, it’s unclear whether they’re testing the interactions on all trials or only hit trials. I’d be interested in what the interactions would be including hit rate.

Thank you for raising this important point of clarification, as it is very important to us that readers understand the analyses that we conducted. As with all the regression analyses in the manuscript, the “effects of maze length on memory” estimated the effect of predictors on hits versus misses, on a trial-by-trial basis. Since our goal was to relate memory for a single item based on trial-by-trial variables, such as reward and proximity, the majority of the analyses we reported in the manuscript examined only the hits for old objects, as the lures were not associated with the necessary trial-by-trial variables. We apologize for the ambiguity and have edited both the supplement and the analyses section of the online method for clarity (supplement p. 29, online method p. 22).

As a behavioral paper, the number of participants in each group (N = 21 in the 15 minute condition and N = 23 in the 24 hour condition) is low. This might be due to the fact that this is perhaps the behavioral data for an imaging study that they’re collecting. But as a stand alone manuscript, the Ns seem low. This is especially concerning given the individual differences mentioned above.

We agree with the reviewer that the number of participants in Experiment 1 was low, which increased the possibility that the reward proximity effect in the 24-hour condition was a false positive. This observation motivated us to collect a second sample of participants in the 24-hour condition (Experiment 2, n = 25), and then we further replicated the effect an additional four times, with larger samples (Experiment 3: n = 32; Experiment 4 – target detection task: n = 27; spatial navigation task: n = 27; working memory condition: n = 23). Together, these results suggest that the reward proximity effect is reliable. To address this issue in the revised manuscript, we have moved the results of Experiment 2 to the main manuscript, emphasizing our motivation for conducting this experiment (see results p. 6).

Reviewer #2 (Remarks to the Author):

In this paper, Kendall Braun et al. investigate how unexpected rewards retroactively influence the formation of memories. Participants were exposed to incidental objects while navigating around a maze. Occasionally, a spatial trajectory was rewarded. Consistent with recent findings in the animal literature, the authors find that reward influences memory strength along a gradient, with a stronger retention of objects experienced just before a reward was experienced, and a weaker retention of objects experienced a longer time ago. This retention depends both on overnight consolidation and the duration of a post-encoding rest period.

This study addresses a timely question using a clever experimental design and the findings will be of interest to other researchers in the community. It is also nice to see that the authors replicate their findings multiple times within the same study. However, I have three major concerns:

1) The authors base the interpretation of their results on a negative effect of proximity on memory in the rewarded mazes. However, looking across all distances, there doesn't seem to be any benefit of reward for memory retention, at least in the 24h condition. Furthermore, in many of the conditions there is not just a negative effect of distance in rewarded mazes, but also a positive effect of distance in the unrewarded mazes (particularly evident in Figure 4). While the positive slope is probably not significant in most conditions, it seems that objects that are further away from the final location of a trajectory are generally better remembered if no reward is encountered compared to a rewarded situation. In other words, it seems that the interpretation that distal events are forgotten if reward is encountered may be more appropriate than the interpretation that proximal events are remembered in rewarded mazes. I am not sure how this can be reconciled with the idea that reward-induced reverse replay strengthens the memory of proximal events. The authors seem to ignore this intriguing observation. Do they have an explanation for this?

We thank the reviewer for these keen observations and insights into the data. There are a number of points worth clarifying here, all of which we now address in the discussion (p. 11).

First, the reviewer's comment about the lack of reward-modulated memory enhancement highlights the important point that the reward proximity effect that we report in six samples reflects a relative difference in memory, across proximity and reward conditions. In that sense, we completely agree that the reward proximity effect is an interaction, not a main effect of reward. Indeed, we did not detect a main effect of reward in any of our datasets in which memory was tested after 24-hours (Experiment 1 – 24-hour condition: $\beta = -0.0037$, $SE = 0.034$, $z = -0.11$, $CI_{95} = [-0.071, 0.061]$, $p = 0.91$; Experiment 2: $\beta = -0.037$, $SE = 0.033$, $z = -1.14$, $CI_{95} = [-0.10, 0.026]$, $p = 0.25$; Experiment 3: $\beta = -0.038$, $SE = 0.028$, $z = -1.33$, $CI_{95} = [-0.094, 0.017]$, $p = 0.18$; Experiment 4 – target condition: $\beta = 0.013$, $SE = 0.029$, $z = 0.46$, $CI_{95} = [-0.042, 0.70]$, $p = 0.64$; navigation condition: $\beta = 0.027$, $SE = 0.029$, $z = 0.94$, $CI_{95} = [-0.029, 0.086]$, $p = 0.35$; working memory condition: $\beta = -0.024$, $SE = 0.032$, $z = -0.76$, $CI_{95} = [-0.087, 0.038]$, $p = 0.45$). To be clearer about this, we have edited our manuscript to describe the retroactive reward proximity effect as a retroactive “prioritization” or “reprioritization”, as this emphasizes the interaction and specifically the graded modulation of memory by proximity to the maze outcome (see title p. 1, abstract p. 2, introduction p. 3, discussion p. 11).

Second, regarding the effect of sequential proximity on memory in the no reward (or low reward) condition, we detected a significant positive slope in four of the six datasets when memory was tested after 24-hours (Experiment 3: $\beta = 0.082$, $SE = 0.038$, $z = 2.17$, $CI_{95} = [0.0084, 0.16]$, $p = 0.030$; Experiment 4 – target condition: $\beta = 0.13$, $SE = 0.041$, $z = 3.14$, $CI_{95} = [-0.050, 0.21]$, $p = 0.0017$; navigation condition: $\beta = 0.11$, $SE = 0.041$, $z = 2.75$, $CI_{95} = [0.034, 0.19]$, $p = 0.0060$; working memory condition: $\beta = 0.15$, $SE = 0.045$, $z = 3.20$, $CI_{95} = [0.058, 0.23]$, $p = 0.0014$) and a trend in the same direction on a fifth experiment (Experiment 2: $\beta = 0.098$, $SE = 0.052$, $z = 1.88$, $CI_{95} = [-0.0043, 0.20]$, $p = 0.061$). We did not detect a significantly positive slope in the first experiment (Experiment 1 – 24-hour condition: $\beta = 0.054$, $SE = 0.045$, $z = 1.21$, $CI_{95} = [-0.032, 0.14]$, $p = 0.23$). We now state these results clearly in the main manuscript (see results p. 6), rather than just in the post-hoc test table in the supplement. Additionally, we directly address this unpredicted finding in the discussion, building on a recent paper by Ambrose, Pfeiffer, & Foster

(2016) to speculate that increased reverse replay following the reward mazes may underpin the negative slope in the rewarded, while relatively stable levels of forward replay may contribute to the positive slope in the no reward condition.

On a related note, does the hit rate suggest that participants perform at chance (at least in the 24h plot)? If so, I am not sure how meaningful the scaling by proximity is. Again, it would be good if the authors could comment on this.

For subsequent recognition memory tasks in which participants make old/new judgments about their memory for items, corrected hit rate (i.e. hits – false alarms) is used to determine if the participants' memory is above chance, as this criterion measures participants ability to discriminate between old and new objects by taking into account participants' response bias. In our experiment, 192 objects were incidentally encoded, with a surprise memory test administered after 24-hour delay. The corrected hit rates for each experiment are well above chance: Experiment 1 – 24-hour condition: 0.23 ± 0.03 , $t(22)=7.97$, $p=0.000000062$; 15-minute condition: 0.34 ± 0.03 , $t(20)=11.44$, $p=0.0000000031$; Experiment 2: 0.18 ± 0.02 , $t(20)=8.86$, $p=0.000000023$; Experiment 3: 0.22 ± 0.02 , $t(31)=11.14$, $p=0.00000000023$; and Experiment 4 – target condition: 0.18 ± 0.02 , $t(26)=10.09$, $p=0.0000000018$; navigation condition: 0.21 ± 0.03 , $t(26)=7.42$, $p=0.000000071$; working memory condition: 0.17 ± 0.02 , $t(22)=7.97$, $p=0.000000063$. These results have been reported in the supplement (p. 25).

Since our goal was to relate memory for a single item based on trial-by-trial variables, such as reward and proximity, the majority of the analyses we report in the manuscript examined only the hits for old objects, as the lures were not associated with the necessary trial-by-trial variables.

2) In their plots, the authors only display the results of the model fit to their data, rather than the data itself. It would be much more informative to see the raw data plotted (at least in addition to the model fits), also because this would help assess whether the linear model is a good fit at all.

We thank the reviewer for this suggestion. We agree that visualizing the data underlying our model is useful for interpreting the data. Therefore, we have included a figure in the supplement that plots the mean memory by proximity and reward. Since we were most interested in examining this plot to see the shape of the underlying data, we have included the data from all of the 24-hour datasets, as this provides the best estimate of the underlying shape (see Figure S1e).

3) Can the effect of reward and proximity on memory be explained by other confounding factors? For example:

We thank the reviewer for identifying these potential confounds. We have carefully conducted the recommended analyses, addressing each concern individually below. Even after we consider these potential issues, the overall pattern of results and our interpretation of these findings remains the same. We have added these analyses to the supplement (see supplement, p. 26).

a. The post-object jitter length, which varies considerably between objects (this is unlikely to vary with distance though, if randomized appropriately).

The jitter was randomized across the reward and no reward mazes. Therefore, when we tested for the effect of time in seconds from the onset of the end outcome ("time to end") on the "steps

to end” sequential proximity measure, we found a very strong relationship between the two variables (Experiment 1 – 24-hour condition: $\beta = 0.16$, SE = 0.0020, $t = 82.43$, $CI_{95} = [0.16, .17]$, $p < 0.0004$; Experiment 2: $\beta = 0.16$, SE = 0.0046, $t = 35.10$, $CI_{95} = [0.15, 0.17]$, $p < 0.0004$; Experiment 3: $\beta = 0.013$, SE = 0.00070, $t = 18.70$, $CI_{95} = [0.012, 0.015]$, $p < 0.0004$; Experiment 4 – target: $\beta = 0.17$, SE = 0.0014, $t = 117.09$, $CI_{95} = [0.16, 0.17]$, $p < 0.004$; navigation: $\beta = 0.16$, SE = 0.0027, $t = 60.31$, $CI_{95} = [0.16, 0.17]$, $p < 0.0004$; working memory: $\beta = 0.16$, SE = 0.0026, $t = 64.29$, $CI_{95} = [0.16, 0.17]$, $p < 0.0004$).

As expected, when we tested for the effect of time in seconds from the onset of the end outcome (“time to end”), we found results that mirror the reward proximity reported in the main manuscript (reward X time to end: Experiment 1 – 24-hour condition: $\beta = -0.11$, SE = 0.038, $z = -3.12$, $CI_{95} = [-0.19, -0.043]$, $p = 0.0018$; Experiment 2: $\beta = -0.14$, SE = 0.033, $z = -4.21$, $CI_{95} = [-.21, -0.075]$, $p < 0.0004$; Experiment 3: $\beta = -0.20$, SE = 0.061, $z = -3.31$, $CI_{95} = [-0.32, -0.081]$, $p < 0.0004$; Experiment 4 – target detection: $\beta = -0.16$, SE = 0.029, $z = -5.79$, $CI_{95} = [-0.23, -0.11]$, $p < 0.0004$; navigation: $\beta = -0.17$, SE = 0.029, $z = -5.73$, $CI_{95} = [-0.23, -0.11]$, $p < 0.0004$; working memory: $\beta = -0.15$, SE = 0.032, $z = -4.62$, $CI_{95} = [-0.21, -0.083]$, $p < 0.0004$).

b. The position of an object on the maze. Participants may have a tendency to explore particular locations, either because they are more salient (e.g. central locations) or because they were previously rewarded. For example, locations in the centre of the maze may be a lot more salient than locations in the surrounding areas. Because of their salience objects at these locations may be easier to remember. But because participants keep navigating to these locations, objects at these locations will also occur closer in time to the reward. The relationship between memory and distance to reward may therefore be incidental.

Thank you for allowing us the opportunity to clarify this important point about the effect of navigational behavior on the reward proximity effect. Critically, during maze exploration, participants did not know the outcome of that maze, and therefore, maze navigational behavior cannot vary between these two conditions. Consequently, navigational behavior cannot confound the reward proximity effect. We have revised the manuscript to make sure that this point is clear (see introduction p. 5, supplement p. 31).

Additionally, the reviewer raises the possibility that inside of the maze may have been more salient, and this salience may have biased memory. Here we operationalized “inside” as the nine squares on the inside of the maze and “outside” as the 16 squares on the perimeter. We did not find that objects inside/outside location modulated memory in any of our experiments (Inside/Outside, coded inside = 1, outside = -1: Experiment 1 – 24-hour condition: $\beta = 0.016$, SE = 0.033, $z = 0.47$, $CI_{95} = [-0.052, 0.081]$, $p = 0.64$; Experiment 2: $\beta = 0.043$, SE = 0.037, $z = 1.16$, $CI_{95} = [-0.029, 0.12]$, $p = 0.25$; Experiment 3: $\beta = -0.023$, SE = 0.032, $z = -0.71$, $CI_{95} = [-0.085, 0.040]$, $p = 0.48$; Experiment 4 – target condition: $\beta = 0.043$, SE = 0.030, $z = 1.43$, $CI_{95} = [-0.015, 0.10]$, $p = 0.17$; navigation condition: $\beta = -0.0060$, SE = 0.030, $z = -0.20$, $CI_{95} = [-0.064, 0.055]$, $p = 0.84$), although in the working memory condition of Experiment 4, we found a trend such that participants were more likely to remember objects from the inside of the maze (working memory condition: $\beta = 0.060$, SE = 0.033, $z = 1.84$, $CI_{95} = [-0.00084, 0.13]$, $p = 0.050$).

Finally, we tested if the reward proximity effect remained, even when we accounted for the inside/outside location of encoding, and we found that the reward proximity effect persisted in every experiment (Reward x Proximity: Experiment 1 – 24-hour condition: $\beta = -0.11$, SE = 0.035, z

= -3.05 $CI_{95} = [-0.17, -0.037]$, $p = 0.004$; Experiment 2: $\beta = -0.13$, $SE = 0.033$, $z = -3.78$ $CI_{95} = [-0.19, -0.061]$, $p < 0.0004$; Experiment 3: $\beta = -0.12$, $SE = 0.027$, $z = -4.46$, $CI_{95} = [-0.17, -0.070]$, $p < 0.0004$; Experiment 4 – target condition: $\beta = -0.14$, $SE = 0.029$, $z = -4.79$ $CI_{95} = [-0.20, -0.084]$, $p < 0.0004$; navigation condition: $\beta = -0.15$, $SE = 0.029$, $z = -5.03$, $CI_{95} = [-0.20, -0.088]$, $p < 0.0004$; working memory condition: $\beta = -0.12$, $SE = 0.032$, $z = -3.95$, $CI_{95} = [-0.19, -0.061]$, $p < 0.0004$). In these models, we did not detect a main effect of inside/outside in any experiment (Inside/Outside: Experiment 1 – 24-hour condition: $\beta = 0.011$, $SE = 0.034$, $z = 0.33$, $CI_{95} = [-0.058, 0.075]$, $p = 0.74$; Experiment 2: $\beta = 0.046$, $SE = 0.038$, $z = 1.21$, $CI_{95} = [-0.028, 0.12]$, $p = 0.23$; Experiment 3: $\beta = -0.022$, $SE = 0.032$, $z = -0.70$, $CI_{95} = [-0.085, 0.041]$, $p = 0.48$; Experiment 4 – target condition: $\beta = 0.043$, $SE = 0.029$, $z = 1.47$, $CI_{95} = [-0.014, 0.10]$, $p = 0.14$; navigation condition: $\beta = -0.0034$, $SE = 0.030$, $z = -0.11$, $CI_{95} = [-0.061, 0.056]$, $p = 0.92$), although again we found a trend such that participants were more likely to remember objects from the inside of the maze (Experiment 4 – working memory condition: $\beta = 0.058$, $SE = 0.033$, $z = 1.77$, $CI_{95} = [-0.0056, 0.12]$, $p = 0.073$). We have added these analyses to the supplement (see p. 31).

Together, these results suggest that the reward proximity effect is not confounded by navigational behavior. This is not surprising, since participants did not know the outcome of any maze during exploration; navigation behavior cannot be modulated by reward and thus cannot explain the reward proximity effect.

c. The number of objects experienced in any given position in space, either before encountering the object of interest or in total (i.e. the number of times a given location has been visited before). This is again related to the reasoning outlined in b).

To answer this question, we tested whether “previous objects in location” affected memory, and whether this effect was different than an overall effect of time in the full task. In Experiments 2, 3 and 4, we found that as the number of objects previously experienced in any maze location increases, the memory for that object decreased (object history by location: Experiment 2: $\beta = -0.035$, $SE = 0.0095$, $z = -3.69$, $CI_{95} = [-0.055, -0.16]$, $p < 0.0004$; Experiment 3: $\beta = -0.037$, $SE = 0.0056$, $z = -6.63$, $CI_{95} = [-0.048, -0.026]$, $p < 0.004$; Experiment 4 – target condition: $\beta = -0.037$, $SE = 0.0093$, $z = -3.99$, $CI_{95} = [-0.056, -0.019]$, $p < 0.0004$; navigation condition: $\beta = -0.025$, $SE = 0.0086$, $z = -2.95$, $CI_{95} = [-0.042, -0.0087]$, $p = 0.0032$; working memory condition: $\beta = -0.052$, $SE = 0.011$, $z = -4.63$, $CI_{95} = [-0.074, -0.031]$, $p < 0.0005$). Additionally, in Experiment 1 – 24-hour condition, we found a trend in the same direction (object history by location: Experiment 1: $\beta = -0.018$, $SE = 0.010$, $z = -1.76$, $CI_{95} = [-0.037, 0.0017]$, $p = 0.066$).

However, “Object History by Location” was confounded with number of trials encoded, and we found strong correlations between “Object History by Location” and “Encoding Trial” in each dataset (Experiment 1 – 24-hour condition: $\beta = 0.049$, $SE = 0.0011$, $t = 43.56$, $CI_{95} = [0.046, 0.051]$, $p < 0.0004$; Experiment 2: $\beta = 0.049$, $SE = 0.0011$, $t = 45.97$, $CI_{95} = [0.047, 0.051]$, $p < 0.0004$; Experiment 3: $\beta = 0.049$, $SE = 0.00090$, $t = 54.29$, $CI_{95} = [0.047, 0.051]$, $p < 0.0004$; Experiment 4 – target: $\beta = 0.047$, $SE = 0.00090$, $t = 52.07$, $CI_{95} = [0.045, 0.049]$, $p < 0.0004$; navigation: $\beta = 2.81$, $SE = 0.076$, $t = 37.22$, $CI_{95} = [0.048, 0.053]$, $p < 0.0004$; working memory: $\beta = 0.050$, $SE = 0.0013$, $t = 37.94$, $CI_{95} = [0.047, 0.052]$, $p < 0.0004$). Therefore, we suspected that the main effect of “Objects History by Location” was actually an effect of trials encoded. To test this, we ran models in which both “Object History by Location” and “Encoding Trial” predicted memory and found that in each model, “Encoding Trial” was a significant predictor of memory, such that as “Encoding Trial” increased, the likelihood of remembering an object decreased (encoding trial: Experiment 1 – 24-

hour condition: $\beta = -0.0026$, $SE = 0.0011$, $z = -2.46$, $CI_{95} = [-0.0046, -0.00060]$, $p = 0.0088$;
Experiment 2: $\beta = -0.0030$, $SE = 0.00098$, $z = -3.14$, $CI_{95} = [-0.0050, -0.0012]$, $p = 0.0008$;
Experiment 3: $\beta = -0.0030$, $SE = 0.00091$, $z = -3.33$, $CI_{95} = [-0.0048, -0.0013]$, $p = 0.0008$;
Experiment 4 – target: $\beta = -0.0037$, $SE = 0.00086$, $z = -4.27$, $CI_{95} = [-0.0053, -0.0020]$, $p < 0.0004$;
working memory: $\beta = -0.0031$, $SE = 0.0010$, $z = -3.00$, $CI_{95} = [-0.0051, -0.00098]$, $p = 0.004$). We did not detect a significant effect of encoding trial in Experiment 4 – navigation condition (encoding trial: $\beta = -0.00079$, $SE = 0.00090$, $z = -0.88$, $CI_{95} = [-0.0025, 0.00099]$, $p = 0.38$). When we considered the effect of object history, we did not detect any significant effects in these models (object history: Experiment 1 – 24-hour condition: $\beta = 0.015$, $SE = 0.016$, $z = 0.98$, $CI_{95} = [-0.016, 0.046]$, $p = 0.34$; Experiment 2: $\beta = 0.0039$, $SE = 0.016$, $z = 0.25$, $CI_{95} = [-0.027, 0.036]$, $p = 0.85$; Experiment 3: $\beta = -0.0044$, $SE = 0.014$, $z = -3.039$, $CI_{95} = [-0.034, 0.024]$, $p = 0.75$; Experiment 4 – target: $\beta = 0.012$, $SE = 0.014$, $z = 0.83$, $CI_{95} = [-0.017, 0.039]$, $p = 0.42$; navigation: $\beta = -0.016$, $SE = 0.013$, $z = -1.23$, $CI_{95} = [-0.042, 0.010]$, $p = 0.21$; working memory: $\beta = -0.012$, $SE = 0.015$, $z = -0.77$, $CI_{95} = [-0.042, 0.017]$, $p = 0.44$).

d. The number of times a reward was encountered at a given object’s location in space (either before an object of interest, or in total). Again related to the reasoning outlined in b).

This is a very interesting question that we had not previously considered. In Experiment 1 and two conditions of Experiment 4, we did not find that reward history by location (i.e. for each object the number of rewards previously experienced in that location) modulates object memory (reward history by location: Experiment 1: $\beta = 0.033$, $SE = 0.061$, $z = 0.55$, $CI_{95} = [-0.091, 0.15]$, $p = 0.61$; Experiment 4 – target: $\beta = -0.037$, $SE = 0.064$, $z = -0.59$, $CI_{95} = [-0.16, 0.087]$, $p = 0.58$; working memory: $\beta = 0.022$, $SE = 0.057$, $z = 0.39$, $CI_{95} = [-0.086, 0.14]$, $p = 0.68$). In Experiments 2, 3 and one condition of Experiment 4, we found a trend such that participants were *less likely* to remember an object, if more rewards had been previously experienced in that location (Experiment 2: $\beta = -0.12$, $SE = 0.064$, $z = -1.94$, $CI_{95} = [-0.25, 0.0090]$, $p = 0.062$; Experiment 3: $\beta = -0.14$, $SE = 0.072$, $z = -1.92$, $CI_{95} = [-0.28, 0.0046]$, $p = 0.058$; navigation: $\beta = -0.098$, $SE = 0.060$, $z = -1.63$, $CI_{95} = [-0.21, 0.021]$, $p = 0.11$). This inconsistent pattern of results suggests that previous reward history by location does not reliably modulate subsequent memory.

Next, we tested if the reward proximity effect remained when we accounted for the previous reward history by location and found that the reward proximity effect persisted in every experiment (reward x proximity: Experiment 1 – 24-hour: $\beta = -0.11$, $SE = 0.035$, $z = -3.09$, $CI_{95} = [-0.18, -0.043]$, $p = 0.0024$; Experiment 2: $\beta = -0.12$, $SE = 0.037$, $z = -3.32$, $CI_{95} = [-0.20, -0.049]$, $p < 0.0004$; Experiment 3: $\beta = -0.12$, $SE = 0.028$, $z = -4.22$, $CI_{95} = [-0.17, -0.062]$, $p < 0.0004$; Experiment 4 – target: $\beta = -0.14$, $SE = 0.029$, $z = -4.79$, $CI_{95} = [-0.20, -0.084]$, $p < 0.0004$; navigation condition: $\beta = -0.15$, $SE = 0.029$, $z = -4.98$, $CI_{95} = [-0.21, -0.092]$, $p < 0.0004$; working memory condition: $\beta = -0.13$, $SE = 0.035$, $z = -3.66$, $CI_{95} = [-0.19, -0.063]$, $p = 0.0008$). In these models, we did not detect a main effect of reward history by location (reward history by location: Experiment 1 – 24-hour: $\beta = 0.039$, $SE = 0.061$, $z = 0.63$, $CI_{95} = [-0.088, 0.16]$, $p = 0.51$; Experiment 2: $\beta = -0.11$, $SE = 0.064$, $z = -1.79$, $CI_{95} = [-0.25, 0.012]$, $p = 0.066$; Experiment 4 – target: $\beta = -0.031$, $SE = 0.065$, $z = -0.49$, $CI_{95} = [-0.16, -0.098]$, $p = 0.61$; navigation: $\beta = -0.084$, $SE = 0.060$, $z = -1.39$, $CI_{95} = [-0.21, 0.033]$, $p = 0.16$; working memory: $\beta = 0.028$, $SE = 0.057$, $z = 0.49$, $CI_{95} = [-0.085, 0.14]$, $p = 0.64$), except in Experiment 3, in which we detected a negative trend, such that as reward history by location increased, memory decreased (reward history by location: $\beta = -0.13$, $SE = 0.073$, $z = -1.76$, $CI_{95} = [-0.27, 0.007]$, $p = 0.078$).

Together these analyses suggest that previous reward history does not confound the reward proximity effect. We have updated the supplement to include these analyses.

Minor comments:

The stimuli are arranged on a spatial maze structure. Temporal succession is therefore not the only conceivable distance measure between objects and rewards. Instead, spatial proximity on the maze may also influence memory. It would be very interesting (albeit not necessary) if the authors found such effects in their data as such a result would nicely add to a range of recent studies investigating cognitive maps in the hippocampus.

Thank you for raising the important question of the role of space in our paradigm. Reviewer #1 raised the same question; please see below:

Thank you for raising the important question of the role of space in our paradigm. Indeed, while the analyses in our original manuscript focused only on sequential proximity, a central feature of our experimental design is that objects are encoded during exploration in a spatial-temporal context, and our original manuscript did not adequately leverage this richness. We have addressed the reviewer's comment in three ways.

First, we added new analyses of a spatial location memory test, which was collected after the recognition memory test in each experiment. As described in the revised manuscript (see results p. 9 and online method p. 17), to test object location memory, an old object was randomly placed in the maze and the participant was instructed to move the object back to the square where the object was originally encoded. We measured the number of steps between the encoded location and the remembered location. Since the maximum possible error varied as a function of the object's original encoding location (i.e. it is possible to have an error of eight steps for an object encoded in the corner, but only four steps away from an object encoded in the center of the maze), to compute an accuracy measure, we scaled each error by the maximum possible error for that particular encoding location. Then we subtracted this score from 1, so that each trial was given a location memory score ranging from 0 (the remembered location was as far away as possible from the original location) to 1 (the location was remembered correctly).

For these analyses, we combined all of the data from all four experiments when memory was tested after 24-hours. First, we found that participants' spatial location memory performance was above chance (mean spatial memory (se) = 0.54 ± 0.00030 , $t(145) = 13.36$, $p < 0.001$). Second, in an analysis that paralleled the results from the recognition memory test, we found that reward retroactively modulated spatial memory for sequentially proximal objects (data from all Experiments 1, 2, 3 and 4, combined: Reward X Proximity: $\beta = -0.0062$, $SE = 0.0022$, $t = -2.83$, $CI_{95} = [-0.010, -0.0018]$, $p = 0.0032$). Third, we found a main effect of reward (data from all four experiments combined: $\beta = 0.0054$, $SE = 0.0022$, $t = 2.47$, $CI_{95} = [0.0011, 0.0095]$, $p = 0.019$), such that spatial location memory for the rewarded mazes was significantly better than the non-rewarded mazes. We have incorporated these results into the manuscript (see results p. 9, figure 5, online method p.17, supplement p. 24, p. 34).

Further, these results were not simply a by-product of the chance-corrected scoring method: we found the same pattern of results when we conduct the analysis using the uncorrected error score (i.e. the number of steps error between the encoded location and the remembered

location). Note that since this measures error in memory performance, better memory is indicated with a lower error (results from Experiments 1-4 combined: reward x proximity: $\beta = 0.037$, SE = 0.014, $t = 2.57$, CI₉₅ = [0.0094, 0.066], $p = 0.0056$; reward: $\beta = -0.040$, SE = 0.014, $t = -2.80$, CI₉₅ = [-0.068, -0.013], $p = 0.0032$; see supplemental results p. 34). We acknowledge that these effects are small, but in combination with the parallel effects in the recognition memory test, these results suggest that spatial memory for a cognitive map is retroactively modulated by reward.

Second, we wanted to explore how rewards retroactively modulated recognition memory for spatially proximal objects. When we tested to see if there was an interaction between reward and *spatially* proximal objects, we found that rewards retroactively modulated memory for spatially proximal objects (or a trend in the same direction) in all four experiments (reward x spatial proximity; Experiment 1 – 24-hour condition: $\beta = -0.086$, SE = 0.032, $z = -2.67$, CI₉₅ = [-0.15, -0.022], $p = 0.011$; Experiment 2: $\beta = -0.067$, SE = 0.035, $z = -1.92$, CI₉₅ = [-0.13, -0.00097], $p = 0.044$; Experiment 3 (new to the revision): $\beta = -0.038$, SE = 0.027, $z = -1.41$, CI₉₅ = [-0.089, 0.014], $p = 0.17$; Experiment 4 (Experiment 3 in the original manuscript) – target condition: $\beta = -0.075$, SE = 0.029, $z = -2.59$, CI₉₅ = [-0.19, -0.063], $p = 0.011$; navigation condition: $\beta = -0.067$, SE = 0.030, $z = -2.26$, CI₉₅ = [-0.13, -0.02], $p = 0.011$; working memory condition: $\beta = -0.056$, SE = 0.032, $z = -1.76$, CI₉₅ = [-0.12, 0.0070], $p = 0.082$). One caveat to this analysis is that in our task participants navigated through the mazes freely, and consequently, sequential proximity and spatial proximity are correlated (spatial proximity ~ sequential proximity; Experiment 1 – 24 hour condition: $\beta = 0.78$, SE = 0.035, $t = 22.44$, CI₉₅ = [0.71, 0.85], $p < 0.0004$, Experiment 2: $\beta = 0.78$, SE = 0.029, $t = 26.87$, CI₉₅ = [0.72, 0.83], $p < 0.0004$; Experiment 3: $\beta = 0.79$, SE = 0.023, $t = 34.24$, CI₉₅ = [0.75, 0.84], $p < 0.0004$, Experiment 4 – target condition: $\beta = 0.72$, SE = 0.035, $t = 20.65$, CI₉₅ = [0.65, 0.79], $p < 0.0004$; navigation condition: $\beta = 0.74$, SE = 0.036, $t = 20.74$, CI₉₅ = [0.67, 0.81], $p < 0.0004$; working memory condition: $\beta = 0.80$, SE = 0.034, $t = 23.30$, CI₉₅ = [0.73, 0.87], $p < 0.0004$). When we controlled for the reward by sequential proximity interaction, we did not detect evidence that reward retroactively modulates memory by spatial proximity (reward x spatial proximity; Experiment 1: $\beta = -0.039$, SE = 0.038, $z = -1.04$, CI₉₅ = [-0.11, 0.034], $p = 0.30$; Experiment 2: $\beta = -0.0030$, SE = 0.040, $z = -0.076$, CI₉₅ = [-0.080, 0.077], $p = 0.94$; Experiment 3: $\beta = 0.038$, SE = 0.032, $z = 1.18$, CI₉₅ = [-0.024, 0.10], $p = 0.24$; Experiment 4 – target condition: $\beta = -0.0094$, SE = 0.033, $z = -0.28$, CI₉₅ = [-0.073, 0.055], $p = 0.78$; navigation condition: $\beta = 0.0079$, SE = 0.034, $z = 0.23$, CI₉₅ = [-0.057, 0.077], $p = 0.84$; working memory condition: $\beta = 0.015$, SE = 0.037, $z = 0.40$, CI₉₅ = [-0.058, 0.085], $p = 0.66$). However, in each of these analyses, we still detected evidence of the reward proximity effect (reward x sequential proximity; Experiment 1: $\beta = -0.086$, SE = 0.038, $z = -2.28$, CI₉₅ = [-0.16, -0.011], $p = 0.024$; Experiment 2: $\beta = -0.12$, SE = 0.040, $z = -3.09$, CI₉₅ = [-0.20, -0.040], $p = 0.0016$; Experiment 3: $\beta = -0.14$, SE = 0.032, $z = -4.43$, CI₉₅ = [-0.20, -0.079], $p < 0.0004$; Experiment 4 – target condition: $\beta = -0.13$, SE = 0.033, $z = -4.03$, CI₉₅ = [-0.20, -0.69], $p < 0.0004$; navigation condition: $\beta = -0.15$, SE = 0.034, $z = -4.44$, CI₉₅ = [-0.22, -0.085], $p < 0.0004$; working memory condition: $\beta = -0.13$, SE = 0.038, $z = -3.44$, CI₉₅ = [-0.21, -0.056], $p < 0.0004$). Together, these results suggest that sequential proximity explains the data better for than spatial proximity, as would be predicted by models of hippocampal replay. We now include these additional analyses in the manuscript (see supplement p. 27).

Third, as described above, we have downplayed our discussion of place cells in the introduction, emphasizing the relevant psychological literature, and instead speculate about the role of place cells as a mechanism for the reward proximity effect in the discussion (see introduction p. 3, discussion p. 11).

With its strong focus on hippocampal coding mechanisms and replay, the introduction reads as if this is a study investigating replay in humans. Maybe make it clearer that study does not directly look for a neural signature of replay, but instead tests behavioural predictions of the phenomenon. The introduction could in general be a bit more concise.

We want to thank the reviewer for their valuable critique of our introduction and framing that dovetails with the comment by Reviewer 1.

We thank the reviewer for raising this important point. Although the experimental question and experimental design were motivated by our desire to link findings from rodent electrophysiology research to human episodic memory, we agree that with this series of behavioral experiments we can only indirectly test the effects of post-encoding mechanisms on memory. We certainly do not wish to inadvertently mislead our readers and have revised the introduction to emphasize the relevant behavioral literature (see introduction p. 3). Further, in the revised manuscript, we now describe the relevant rodent electrophysiology research in the discussion (see p. 10), where we are careful to point out the speculative nature of these links between this research and our own results. We also modified the abstract accordingly.

Additionally, we have clarified our language throughout the manuscript to ensure that we are communicating our findings precisely (see introduction p. 3 and discussion p. 10). We have removed the sentence “these results are consistent with neuropsychological findings in rodents demonstrating that replay of experiences immediately following encoding and during sleep is important for subsequent choice behavior” (see discussion p. 10).

How were participants instructed? Did they receive any information as to where a reward would be found? Could they have been under the impression that (a) rewards are distributed randomly and it is therefore advisable to keep searching for rewards in different locations on different trials or that (b) rewards are more likely to be found in certain parts of the maze?

We thank the reviewer for noting the importance of the participants’ instructions. We completely agree and, in addition to adding including the instructions below, have now added the verbatim instructions to the supplement (p. 24). As you can see, we deliberately did not provide participants with any information as to where a reward would be found, as we did not want to bias their navigational behavior.

For the Phase 1: maze exploration (incidental encoding) task, participants received the following instructions, presented on the computer screen one or two sentences at a time and read out loud by the experimenter:

Welcome to the Matrix Game! In this game, you will explore a series of mazes. Your goal is to find the gold coin hidden within each maze. You will be paid \$1 bonus for every gold coin that you find. The maze is a 5x5 grid of grey squares. A black frame will indicate where you are in the maze. You can navigate to the space above, below, to the left or right by choosing the corresponding arrow key. You may make a move any time all of the maze spaces are grey. You will have two seconds to make a choice. In each maze you will only get to make a limited number of moves, so if you do not respond in time, you will not have the opportunity to explore as many spaces. The number of moves allowed will vary by maze. After each maze, there will be a short

break before the following maze. You should rest during this time. The next maze will begin automatically.

Additionally, in the supplement, we include the instructions for Phase 2: surprise recognition memory test and Phase 3: surprise spatial location memory test (see p. 24).

Reviewer #3 (Remarks to the Author):

Braun and colleagues report a series of behavioral studies that look at the retroactive influence of reward on memory. In the paradigm they used, subjects explored a “maze” (or grid), uncovering objects at each location until they either discovered a reward (a coin) or the trial ended. Of critical interest was whether finding the reward (coin) influenced memory for the preceding objects in a temporally-graded manner, with preferential memory enhancement for the items closest (in time) to the reward. Indeed, across several studies (essentially 5 independent variants), they find a highly consistent pattern of results where memory for the objects varies according to reward AND proximity to the reward (i.e., there is an interaction). This interaction is driven by a negative slope relating position to memory in the reward condition (worse memory for objects further from the reward) and a generally positive slope for the non-reward condition (relatively better memory for objects further away from the reward). However, this pattern of results was only observed after a 24-hour delay and it also increased as a function of the amount of time that followed each reward cue (i.e., the inter-trial interval).

The results are very intriguing and potentially establish some novel, important results. The way the paper is framed is compelling and nicely motivates the ideas. Moreover, the multiple replications make a strong case that the effect is “real.” The sensitivity of the effect to delay and to the post-reward time interval are interesting results, as well. That said, I am less convinced that the actual mechanism driving the results has been worked out. The results are intriguing enough that this may not necessarily be a fatal flaw, but I think there are some important points that need to be addressed/considered. Additionally, the paper could benefit from some editing as I think the presentation of results is a bit confusing.

Major Comments

1. Lack of reward enhancement. To me, the “elephant in the room” is that there is essentially no evidence that reward had an overall positive influence on memory. If anything, the only hint of an effect was in the 15 minute delay condition. But in the 24 hour delay conditions, there is no evidence for a main effect of reward. Thus, claims of “enhancement of memory by reward” (as appears in the title and elsewhere in the paper) are misleading. Likewise, there are statements such as “reward retroactively and selectively enhanced memory.” Clearly, there are differences between the reward condition and the no reward condition, but I am not sure it can be called enhancement. Even if specifically considering the single item just before the reward cue (or end of the trial), I don’t believe the authors actually report whether memory for this individual item is significantly enhanced. I am assuming it is, but even if it is, this enhancement seems to come at the expense of memory for other items. Thus, it is more like a reprioritization that is induced by reward. Whatever enhancement there might be, there are offsetting costs. The lack of an overall effect of reward was not considered in the Discussion, but I think this point warrants explicit discussion

Thank you for raising this important issue. Reviewer #2 was also concerned about the main effect of reward; please see our response copied below.

We thank the reviewer for these keen observations and insights into the data. There are a number of points worth clarifying here, all of which we now address in the discussion (p. 11).

First, the reviewer's comment about the lack of reward-modulated memory enhancement highlights the important point that the reward proximity effect that we report in six samples reflects a relative difference in memory, across proximity and reward conditions. In that sense, we completely agree that the reward proximity effect is an interaction, not a main effect of reward. Indeed, we did not detect a main effect of reward in any of our datasets in which memory was tested after 24-hours (Experiment 1 – 24-hour condition: $\beta = -0.0037$, $SE = 0.034$, $z = -0.11$, $CI_{95} = [-0.071, 0.061]$, $p = 0.91$; Experiment 2: $\beta = -0.037$, $SE = 0.033$, $z = -1.14$, $CI_{95} = [-0.10, 0.026]$, $p = 0.25$; Experiment 3: $\beta = -0.038$, $SE = 0.028$, $z = -1.33$, $CI_{95} = [-0.094, 0.017]$, $p = 0.18$; Experiment 4 – target condition: $\beta = 0.013$, $SE = 0.029$, $z = 0.46$, $CI_{95} = [-0.042, 0.70]$, $p = 0.64$; navigation condition: $\beta = 0.027$, $SE = 0.029$, $z = 0.94$, $CI_{95} = [-0.029, 0.086]$, $p = 0.35$; working memory condition: $\beta = -0.024$, $SE = 0.032$, $z = -0.76$, $CI_{95} = [-0.087, 0.038]$, $p = 0.45$). To be clearer about this, we have edited our manuscript to describe the retroactive reward proximity effect as a retroactive “prioritization” or “reprioritization”, as this emphasizes the interaction and specifically the graded modulation of memory by proximity to the maze outcome (see title p. 1, abstract p. 2, introduction p. 3, discussion p. 11).

Additionally, to further describe the selective prioritization, we now report the effect of reward on memory for the object immediately prior to the maze outcome. We find that memory for the rewarded object is significantly greater than the non-rewarded object in most of the datasets (Experiment 1 – 24 hour condition: $\beta = 0.52$, $SE = 0.20$, $z = 2.55$, $CI_{95} = [0.12, 0.91]$, $p = 0.013$; Experiment 3: $\beta = 0.36$, $SE = 0.16$, $z = 2.27$, $CI_{95} = [0.061, 0.67]$, $p = 0.015$; Experiment 4 – target condition: $\beta = 0.50$, $SE = 0.17$, $z = 2.98$, $CI_{95} = [0.17, 0.84]$, $p = 0.0072$; navigation condition: $\beta = 0.44$, $SE = 0.17$, $z = 2.60$, $CI_{95} = [0.11, 0.77]$, $p = 0.010$). We did not detect an effect of reward for memory on the first step in either Experiment 2, ($\beta = 0.12$, $SE = 0.20$, $z = 0.61$, $CI_{95} = [-0.27, 0.52]$, $p = 0.57$) or Experiment 4 – working memory condition ($\beta = 0.26$, $SE = 0.19$, $z = 1.39$, $CI_{95} = [-0.12, 0.62]$, $p = 0.16$), although the effects are in same direction. We have added these results to the supplement (see p. 26).

Further, we now include a plot of mean memory by reward and proximity (see Figure S1e) and with this we report that closest to the maze outcome objects from the rewarded maze are remembered significantly better than objects from the non-rewarded mazes and the reverse pattern for objects that are far away from the maze outcome (all 24-hour data sets combined: main effect of reward for objects 1, 2, or 3 steps from maze outcome: $\beta = 0.26$, $SE = 0.041$, $z = 6.29$, $p < 0.0004$; 4, 5, or 6 steps: $\beta = 0.0036$, $SE = 0.043$, $z = -0.084$, $p = 0.93$; 7, 8, or 9 steps: $\beta = -0.30$, $SE = 0.052$, $z = -5.78$, $p < 0.0004$; 10, 11, or 12 steps: $\beta = -0.36$, $SE = 0.082$, $z = -4.33$, $p < 0.0004$; 13, 14, or 15 steps: $\beta = -0.45$, $SE = 0.14$, $z = -3.12$, $p = 0.0018$). We report these statistics in the figure caption.

1a. Another example related to the reward enhancement claims: In Fig1a, it states “In the 24-hour condition, reward retroactively modulated memory, such that participants were more likely to remember objects that were more proximal to the reward.” But the statistic that is reported here is actually the interaction between reward and proximity. To say that reward modulated memory such

that participants were more likely to remember objects that were proximal to the reward, they should report whether the slope for the reward condition (alone) was significant. They don't seem to report this. While they do show the mean beta for the reward condition in the inset, the error bars here are from the interaction, so it's not possible to assess whether the beta is significantly negative for the reward condition alone.

Thank you for bringing this statistical omission to our attention. In the original manuscript, we reported the betas and standard errors for the reward and no reward mazes in the supplement, but you are absolutely correct that we need to report these statistics in the main manuscript to interpret the data as we do. These statistics have been added to the results section and figure captions (see results p. 6).

2. Is the reward effect actually a reward effect? Again, there is clearly a difference between the reward and no reward conditions. But, I am wondering whether this effect is due to reward, per se. I realize this may sound like a subtle point, but the reward and no reward conditions are not perfectly matched in that the reward trials involved presentation of the coin at a location ON THE GRID (and necessarily spatially adjacent to the immediately preceding object), whereas the no reward trials involved presentation of a message ABOVE THE GRID (and this message was of course not spatially adjacent to the immediately preceding object). This had me wondering whether the "MAZE OVER" screen might have functioned to draw subjects' attention away from the grid and disrupted memory for the preceding items. As I note above, the reward effect appears to be driven both by the negative slope for the reward condition and a positive slope for the no reward condition. This raises the question of why there was a positive slope for the no reward condition. Perhaps this simply reflects buildup of interference across the list, but it could also reflect a disruption from the "MAZE OVER" screen. The most obvious solution to this issue would be to better match the reward and no reward trials such that for the no reward trials a "worthless" coin appeared in precisely the same spot that the reward coin would appear. But, with the current procedure, I am not entirely convinced that the reward effect is related to reward, per se. The alternative account would be something about interference/distraction. Of course, even with this alternative account, there are still the interesting facts that the influence emerges over time (24 hour delay) and that it is sensitive to the inter-trial interval.

Thank you for raising this important issue. Reviewer #1 brought up a similar concern; please see our response below.

Thank you for identifying this concern. To address it, we ran a new experiment ("Experiment 3", see results p. 6, p. 8, Figure S1c, Figure S3b,d, online method p. 19). We opted to isolate the effect of reward as conservatively as possible by comparing low vs. high rewards, removing the no reward ("maze over") condition altogether. In this new experiment, participants navigated through a series of mazes that ended either in a dime (worth \$0.10 bonus) or the same gold coin reward used previously (worth \$1 bonus), so that the maze outcome always appeared as a coin within the maze.

The results of Experiment 3 replicated the reward proximity finding described in the original manuscript: participants had better memory for the objects leading up to the gold coin relative to objects leading up to the dime (reward x proximity: $\beta = -0.12$, $SE = 0.027$, $z = 4.48$, $CI_{95} = [-0.17, -0.071]$, $p < 0.0004$). We also found, again, that the duration of the post-encoding rest break interacted with the reward proximity effect, such that the reward proximity effect was stronger

when the post-encoding rest break was longer (reward x proximity x rest duration: $\beta = -0.018$, SE = 0.035, $z = -5.06$, $CI_{95} = [-0.25, -0.11]$, $p < 0.0004$) and that the model including rest duration explained the data significantly better than the simpler model omitting rest duration ($\chi^2(8) = 38.90$, $p = 0.000051$), replicating the effects reported originally. Together, these results suggest that the pattern of results reported in our original manuscript cannot be explained by differences related to the maze outcome being presented in the maze, but instead are due to the value of the reward.

3. The fact that there is a positive slope for the no reward condition also seems to go against the idea in the Introduction that replay occurs in reverse order after each event. If this were true, then we should expect a negative slope for BOTH the reward and no reward conditions but that the slope would simply be steeper and/or shifted up for the reward condition.

Thank you for raising the issue of the positive slope for the no reward mazes, as we did not sufficiently address the issue in the original manuscript. Reviewer #2 raised similar questions; please see our response below.

Regarding the effect of sequential proximity on memory in the no reward (or low reward) condition, we detected a significant positive slope in four of the six datasets when memory was tested after 24-hours (Experiment 3: $\beta = 0.082$, SE = 0.038, $z = 2.17$, $CI_{95} = [0.0084, 0.16]$, $p = 0.030$; Experiment 4 – target condition: $\beta = 0.13$, SE = 0.041, $z = 3.14$, $CI_{95} = [-0.050, 0.21]$, $p = 0.0017$; navigation condition: $\beta = 0.11$, SE = 0.041, $z = 2.75$, $CI_{95} = [0.034, 0.19]$, $p = 0.0060$; working memory condition: $\beta = 0.15$, SE = 0.045, $z = 3.20$, $CI_{95} = [0.058, 0.23]$, $p = 0.0014$) and a trend in the same direction on a fifth experiment (Experiment 2: $\beta = 0.098$, SE = 0.052, $z = 1.88$, $CI_{95} = [-0.0043, 0.20]$, $p = 0.061$). We did not detect a significantly positive slope in the first experiment (Experiment 1 – 24-hour condition: $\beta = 0.054$, SE = 0.045, $z = 1.21$, $CI_{95} = [-0.032, 0.14]$, $p = 0.23$). We now state these results clearly in the main manuscript (see results p. 6), rather than just in the post-hoc test table in the supplement. Additionally, we directly address this unpredicted finding in the discussion, building on a recent paper by Ambrose, Pfeiffer, & Foster (2016) to speculate that increased reverse replay following the reward mazes may underpin the negative slope in the rewarded, while relatively stable levels of forward replay may contribute to the positive slope in the no reward condition.

4. All of the effects relating temporal position to memory are based on linear relationships. Are the effects actually linear? Relatedly, I am wondering how much of the effect is driven by memory for the single item prior to the reward (or “MAZE OVER” cue)? It would be helpful to see the data plotted as a function of number of steps from the end point (i.e., n-1, n-2, etc.). As is, we only see the fitted lines, not any of the actual mean data.

Thank you for raising these issues regarding the mean data underlying the models we report.

First, although we find that the object immediately preceding the maze outcome is remembered better for rewarded maze compared to no (or low) reward maze, we do not believe that the reward proximity effect is being driven solely by differences in memory for the object immediately preceding the maze (Experiment 1 – 24 hour condition: $\beta = 0.52$, SE = 0.20, $z = 2.55$, $CI_{95} = [0.12, 0.91]$, $p = 0.013$; Experiment 3: $\beta = 0.36$, SE = 0.16, $z = 2.27$, $CI_{95} = [0.061, 0.67]$, $p = 0.015$; Experiment 4 – target condition: $\beta = 0.50$, SE = 0.17, $z = 2.98$, $CI_{95} = [0.17, 0.84]$, $p = 0.0072$; navigation condition: $\beta = 0.44$, SE = 0.17, $z = 2.60$, $CI_{95} = [0.11, 0.77]$, $p = 0.010$). We did not detect an effect of

reward for memory on the first step in either Experiment 2, ($\beta=0.12$, $SE=0.20$, $z=0.61$ $CI_{95}=[-0.27, 0.52]$, $p=0.57$) or Experiment 4 – working memory condition ($\beta=0.26$, $SE=0.19$, $z=1.39$, $CI_{95}=[-0.12, 0.62]$, $p=0.16$), although the effects are in same direction (see Supplement p. 26). In addition to finding that memory for rewarded mazes is improved for objects close to the reward, we also find that memory for objects that are far away from the maze outcome are remembered better in no reward (or low reward) mazes (all 24-hour data sets combined: main effect of reward for objects 1, 2, or 3 steps from maze outcome: $\beta = 0.26$, $SE = 0.041$, $z = 6.29$, $p < 0.0004$; 4, 5, or 6 steps: $\beta = 0.0036$, $SE = 0.043$, $z = -0.084$, $p = 0.93$; 7, 8, or 9 steps: $\beta = -0.30$, $SE = 0.052$, $z = -5.78$, $p < 0.0004$; 10, 11, or 12 steps: $\beta = -0.36$, $SE = 0.082$, $z = -4.33$, $p < 0.0004$; 13, 14, or 15 steps: $\beta = -0.45$, $SE = 0.14$, $z = -3.12$, $p = 0.0018$) (see Figure S1 caption). Additionally, we now include a plot of mean memory by reward and proximity (see Figure S1e).

Consistent with this, when we log transform the proximity measure to create a curvilinear predictor that would fit the data better if the reward proximity effects were being driven only by memory for the objects closest to the maze outcome and model both mean-centered “steps to end” interacted with reward and “logged steps to end” interacted with reward, we found that the reward by proximity interaction reported in the manuscript emerges (all 24-hour datasets combined: reward X proximity: $\beta = -0.15$, $SE = 0.034$, $z = -4.53$, $p < 0.004$), but we did not detect an interaction between the log transformed proximity and reward (all 24-hour datasets combined: reward X log proximity: $\beta = 0.027$, $SE = 0.034$, $z = 0.81$, $p = 0.42$). We have added this analysis to the supplement (see p. 27).

Therefore, while we do not take a strong position that the effects of reward on memory are strictly linear, we believe that a linear model is an appropriate model to use to characterize the reward proximity effects.

5. The fact that the reward effect requires 25 seconds of inter-trial interval is interesting, but I also find these results surprising. The argument is that events are replayed in the delay condition, but isn't 15 seconds (the shortest interval) enough time to at least replay the last couple of objects? If it were a difference of 1s vs. 10s, that would be more intuitively reasonable to me, but 15 seconds seems like an awfully long time (i.e., enough time for replay). I realize the data are the data, but I am curious if the authors have any intuitions about the timing and/or how they settled on these numbers.

Thank you for drawing our attention to the important issue of the effect of the post-encoding rest duration on memory. Reviewer #1 was also concerned about the interpretation of rest effects; please see our response below.

Thank you for giving us an opportunity to explain our motivation for choosing the duration of the rest intervals. When we designed this experiment, we were motivated by evidence emerging from the rodent literature that linked hippocampal replay to memory and endeavored to develop a human experiment that paralleled these experiments in a reasonably close way. We initially chose a rest duration that loosely approximated the length of an inter-trial interval in such experiments, and then, as we plan to run an fMRI experiment to measure brain activity during these intervals, we increased the time windows to acquire more fMRI data during this critical time period. In piloting, we found that these time windows effectively modulated memory, and consequently, we have not changed them in the ensuing experiments. We have added an explanation for how we chose these time durations in the discussion (see discussion p. 11).

In the discussion (see p. 11), we also now explicitly address the point the reviewer raises, which is that it is not entirely clear from the replay data in rodents why a longer interval would be needed. Because hippocampal replay in rodents and post-encoding reactivation in human neuroimaging persists for an extended period of time following encoding, we speculate that the amount of replay, as opposed to a single instance of replay, may be important.

6. Related to the above point: if the putative replay mechanism is not disrupted by the distractor tasks (target detection, working memory, navigation), then why would/should it be disrupted by the onset of the next trial? In other words, it almost seems like the replay should occur no matter what, meaning that the inter-trial interval length would be irrelevant.

We thank the reviewer for raising such an interesting question about the mechanism supporting the retroactive reward proximity effect. Working under the assumption that the hippocampus is necessary for this reward proximity memory effect, since rest, the target detection task, the navigation task are all tasks that are not hippocampus-dependent (i.e. a person with hippocampal lesions would be capable of completing the task), we are not surprised that they do not interfere with the post-encoding retroactive modulation. Additionally, *if* hippocampal replay is the mechanism that underlies the retroactive reward proximity effect, since replay does not occur while rodents are in theta rhythm during maze navigation, one possibility is that navigating the maze in goal directed manner (as opposed to navigating a simpler stimulus-response type navigation in the navigation condition of Experiment 4) induces a theta-like response that promotes encoding and inhibits replay.

7. The way experiments 1 and 2 are split between the main text and the supplement is confusing. The main text refers you to the supplementary figures for the results of Experiment 2, but then you need to go to the methods to figure out what Experiment 2 actually involved. Since experiment 2 was a “straight replication” why not just include the replication statistic in the main Results section? Also having the figures for Experiments 1 and 2 split across the main text and Supplement makes it more of a hassle to compare.

We wish to thank the reviewer for this excellent suggestion that has improved the readability of our manuscript. In the revised manuscript, we have moved Experiment #2 from the Supplement to the main manuscript (see p. 6).

Minor Comments

8. Statistics are not always reported in the main text and in some cases, it is hard to follow which experiment the data are coming from. For example, for the section describing the reward proximity effect as a function of rest duration, I initially thought this was data from a new experiment. The figure caption does say “experiment 1” but the main text does not list an experiment number or otherwise make it clear that the data are from experiment 1.

Again, we want to thank the reviewer for this helpful suggestion that has improved the readability of our manuscript. We have included the statistics in the main text of the manuscript (beginning p. 6). Additionally, we have clearly labeled Experiment 1 in the main text (see p. 6).

9. In Supplementary Figure 3b, are the betas shown in the inset correct? They appear to be swapped or otherwise incorrect in that the mean beta for the reward condition is shown as negative but the slope of the line in the main part of the figure is positive.

Thank you for the astute observation. In the main figures we plot the predicted values from the models; however, to calculate the beta values for the inset, we re-run the model for each condition (i.e. 15 seconds, 20 seconds, 25 seconds) and use the betas from the subset model for the inset. Consequently, occasionally, these two ways of visualizing the data do not completely align.

10. Please elaborate on why subjects were excluded for “non-compliance.”

Thank you for raising this point of clarification. For the subject counts, we included all of the participants consented into the experiments. In Experiment 2, we excluded two participants for non-compliance: one participant was excluded because the participant did not show up to the second session (i.e. the memory test); and one participant was excluded because the participant was caught using his phone during the experiment and refused to leave the phone outside of the testing room. We have added this information to the online methods section, so that our data exclusion process is more transparent (see p. 18).

11. On p. 9 the text says “These results suggest that the short time window following encoding is a prerequisite for the subsequent differential consolidation of these memory traces.” I found this sentence confusing. At first, I thought the point of the sentence was to emphasize the short time window CONDITION (i.e., the 15 second condition) but I believe instead the authors simply mean that the delay period following each maze was important.

Thank you for letting us know that our language was ambiguous. You are correct that we intended to communicate that the rest break following each maze is important, so we have edited the sentence to read, “These results suggest that the rest breaks immediately following encoding are critical for the differential consolidation of memory that emerges after consolidation” (see results p. 8).

REVIEWERS' COMMENTS:

Reviewer #2 (Remarks to the Author):

The authors have addressed all my concerns.

Reviewer #3 (Remarks to the Author):

The authors have thoughtfully responded to the comments I (and the other Reviewers) raised. In particular, the results of the new Experiment 3 are very compelling, as they replicate the critical finding (strongly) while eliminating a subtle confound that I (and another Reviewer) had noted. While there are some points that aren't--and can't be--fully resolved (e.g., are these effects driven by reactivation, per se), I find the results to be very intriguing and appreciate the multiple replications.